# Regretful Decisions under Label Noise

**Sujay Nagaraj**
University of Toronto

**Yang Liu**
UC Santa Cruz

**Flavio P. Calmon**
Harvard SEAS

**Berk Ustun**
UC San Diego

## Abstract

Machine learning models are routinely used to support decisions that affect individuals – be it to screen a patient for a serious illness or to gauge their response to treatment. In these tasks, we are limited to learning models from datasets with *noisy labels*. In this paper, we study the instance-level impact of learning under label noise. We introduce a notion of *regret* for this regime, which measures the number of unforeseen mistakes due to noisy labels. We show that standard approaches to learning under label noise can return models that perform well at a population-level while subjecting individuals to a *lottery of mistakes*. We present a versatile approach to estimate the likelihood of mistakes at the individual-level from a noisy dataset by training models over plausible realizations of datasets without label noise. This is supported by a comprehensive empirical study of label noise in clinical prediction tasks. Our results reveal how failure to anticipate mistakes can compromise model reliability and adoption – we demonstrate how we can address these challenges by anticipating and avoiding regretful decisions.

## 1 Introduction

Machine learning models are routinely used to support or automate decisions that affect individuals – be it to screen a patient for a mental illness [52], or assess their risk for an adverse treatment response [3]. In such tasks, we train models with labels that reflect noisy observations of the true outcome we wish to predict. In practice, such noise may arise due to measurement error [e.g., 23, 39], human annotation [30], or inherent ambiguity [39]. In all these cases, label noise can have detrimental effects on model performance [11]. Over the past decade, these issues have led to extensive work on *learning from noisy datasets* [see e.g., 11, 32, 40, 44, 49]. As a result, we have developed foundational results that characterize when label noise can be ignored and algorithms to mitigate its detrimental effects.

By and large, this work has focused on the impact of label noise at the population-level. In contrast, studying the effects of label noise at the instance-level has received limited attention. This oversight reflects the fact that we cannot provide meaningful guarantees on individual predictions under label noise [32]. In a best-case scenario, where we have perfectly specified distributional assumptions on label noise, *we can learn a model that performs well on average, but we cannot identify where it makes mistakes*; as a result, individuals are subject to a "lottery of mistakes."

These effects undermine the utility of models in major real-world applications, as label noise arises in many settings where models are used to support or automate individual decisions [see, e.g., 55, for a meta-review of 72 cases in medicine]. In medical decision support tasks, our inability to identify mistakes can lead to overreliance, where physicians rely on predictions that may be incorrect [7, 29]. In automation tasks, our failure to assess the confidence of predictions can prevent us from reaping broader benefits – e.g., by abstention [10, 18].

In this work, we study how label noise affects individual predictions. Our motivation stems from the fact that, even if we cannot fully resolve the effects of label noise at the instance-level, we can mitigate harm by anticipating regretful predictions through uncertainty quantification. To this end, our main contributions are:

1. We introduce a notion of *regret* for learning from noisy datasets, capturing how label uncertainty affects individual predictions. We show that learning under label noise leads to inevitable regret, characterizing key limitations in a wide class of methods for learning from label noise.

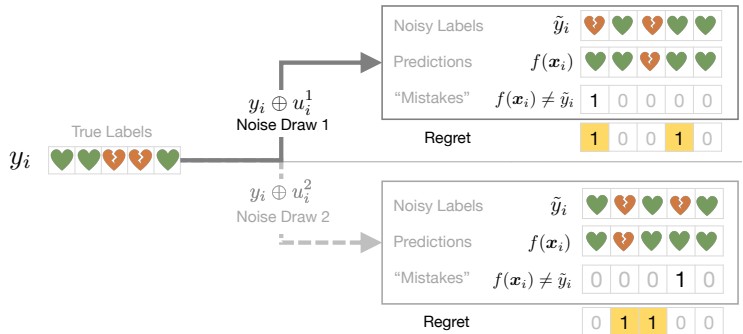

**Figure 1:** Datasets with noisy labels only contain a single draw of label noise. In such settings, we can learn a model that performs well at a population-level but cannot anticipate its mistakes. We characterize the number of individuals who are subjected to a lottery of mistakes in terms of *regret* – i.e., the difference between anticipated mistakes and actual mistakes. Here, we show a stylized classification task with 5 points, where each point with a positive label may be flipped with a probability of 30%. In this case, 4 points are subject to a lottery of mistakes and our model assigns regretful predictions to 2 points, highlighted in yellow.

2. We develop a method to flag regretful predictions by training models on plausible realizations of a clean dataset. Our approach can measure the sensitivity of individual predictions under label noise and incorporates common noise assumptions while controlling for plausibility.

3. We conduct a comprehensive empirical study on clinical prediction tasks. Our findings highlight the instance-level impact of label noise, and we demonstrate how our approach can support safer inference by flagging potential mistakes.

**Related Work**   Our work is related to a stream of research on learning from noisy labels. We focus on applications where we cannot resolve label noise by acquiring clean labels [see e.g., 11, 49, for surveys]. Many methods learn models by hedging for uncertainty in labels [33, 40, 44]. As we show in Section 2, such approaches are robust to label noise at a population-level while subjecting individuals to a lottery of mistakes. Our work highlights the limitations of this regime. In this sense, our results complement the work of Oyen et al. [43], who characterize the lack of robustness to label noise under general distributional assumptions.

We propose to mitigate these issues through a principled approach to uncertainty quantification. Our approach relates to recent work on model multiplicity, which shows how changes in the machine learning pipeline can produce models that assign conflicting predictions [see e.g., 4, 8, 20, 35, 38, 42, 53] and lead to downstream effects on fairness, explanations, and recourse [5, 17, 27, 36]. With respect to the literature on label noise, our approach is similar to the work of Reed et al. [47], who propose training an ensemble of deep neural networks by sampling alternative realizations of clean labels. In contrast, our procedure samples plausible realizations of clean labels and retrains plausible models to quantify uncertainty at an individual-level rather than predict.

## 2   PRELIMINARIES

We consider a classification task where we wish to learn a model $f : \mathcal{X} \to \mathcal{Y}$ to predict a label $y \in \mathcal{Y}$ from a feature vector $\boldsymbol{x} \in \mathcal{X} \subseteq \mathbb{R}^d$. In a standard regime, we would be given a dataset $\mathcal{D} = \{(\boldsymbol{x}_i, y_i)\}_{i=1}^n$ where each $(\boldsymbol{x}_i, y_i)$ is drawn from a joint distribution of random variables, $X$ and $Y$. Given the dataset, we would learn a model that performs well in deployment – i.e., that minimizes the *true risk* $R(f) := \mathbb{E}_{X,Y}[\mathbb{I}[f(X) \neq Y]]$.

We consider a variant of this task where we learn a model from a *noisy dataset* $\tilde{\mathcal{D}} = \{(\boldsymbol{x}_i, \tilde{y}_i)\}_{i=1}^n$, where each *noisy label* $\tilde{y}_i$ represents a potentially corrupted *true label* $y_i$. In what follows, we refer to this corruption as a *flip* and denote it $u_i := \mathbb{I}[y_i \neq \tilde{y}_i]$. Given the flip $u_i$, we can express noisy labels in terms of true labels as $\tilde{y}_i := y_i \oplus u_i$ and vice-versa as $y_i := \tilde{y}_i \oplus u_i$. Here, $a \oplus b := a + b - 2ab$ is the XOR operator. Given a noisy dataset, we represent all flips as a vector called the *noise draw*.

**Definition 1.** Given a binary classification task with $n$ examples, the *noise draw* $\boldsymbol{u} = [u_1, \ldots, u_n] \subseteq \{0,1\}^n$ is a realization of $n$ random variables $[U_1, \ldots, U_n] \subseteq \{0,1\}^n$.

Given an example $(\boldsymbol{x}_i, y_i)$, each flip $u_i$ is drawn from a Bernoulli distribution with parameters $p_{u|y_i, \boldsymbol{x}_i} := \Pr(U_i = 1 \mid X = \boldsymbol{x}_i, Y = y_i)$. Thus, the noise is generated by the random process:

$$U_i \sim \mathsf{Bernouilli}(p_{u|y_i, \boldsymbol{x}_i})$$
$$\tilde{y}_i = y_i \oplus U_i$$

In what follows, we assume that the values $p_{u|y_i, \boldsymbol{x}_i}$ are determined by a generic *noise model* that can take on different forms – e.g., uniform, class-level, or instance-level as shown in Table 1. We write $p_u$ instead of $p_{u|y_i, \boldsymbol{x}_i}$ when conditioning terms are irrelevant or clear from context. We assume that the model is correctly specified and that $p_u < 0.5$ for all points to ensure there are more clean than noisy labels [c.f., 1, 40, 44].

Given a noisy dataset, we denote the noise draw over all instances as the *true draw* $\boldsymbol{u}^{\mathrm{true}} := [u_1^{\mathrm{true}}, \ldots, u_n^{\mathrm{true}}]$. In practice, the true draw $\boldsymbol{u}^{\mathrm{true}}$ is fixed but unknown. From the practitioner's perspective, $\boldsymbol{u}^{\mathrm{true}}$ could be any realization of random variables $U$. If they knew $\boldsymbol{u}^{\mathrm{true}}$, they could recover the true labels as $y_i = \tilde{y}_i \oplus u_i^{\mathrm{true}}$ and learn without label noise. As this is infeasible, given the noise model and a set of priors, practitioners can estimate the posterior noise model $q_{u|\tilde{y}_i, \boldsymbol{x}_i} := \Pr(U_i = 1 \mid X = \boldsymbol{x}_i, \tilde{Y} = \tilde{y}_i)$ to infer clean labels from observed noisy labels.

| Noise Type | PGM | Noise Model | Posterior Model | Inference Requirements | Sample Use Case |
|---|---|---|---|---|---|
| Uniform | $U$ | $p_u = \Pr(U = 1)$ | $q_u = \Pr(U = 1)$ | – | Screening tests with a fixed failure rate [e.g., COVID rapid tests 2]. |
| Class-Level | $Y$ $U$ | $p_{u|y} = \Pr(U = 1 \mid Y = y)$ | $q_{u|\tilde{y}} = \Pr\left(U = 1 \mid \tilde{Y} = \tilde{y}\right)$ | $\pi_y = \Pr(Y = y)$ | Chest X-ray diagnosis where label noise $\tilde{Y}$ changes based on the disease $Y$ [e.g., pneumonia vs COVID 15]. |
| Group-Level | $Y$ $U$ $G$ | $p_{u|y,g} = \Pr(U = 1 \mid Y = y, G = g)$ | $q_{u|\tilde{y},g} = \Pr\left(U = 1 \mid \tilde{Y} = \tilde{y}, G = g\right)$ | $\pi_{y,g} = \Pr(Y = y \mid G = g)$ | Diagnostic tasks where the incidence of label noise changes across subpopulations [e.g., racial bias in diagnosis 14, 51]. |
| Instance-Level | $Y$ $U$ $X$ | $p_{u|y,\boldsymbol{x}} = \Pr(U = 1 \mid Y = y, X = \boldsymbol{x})$ | $q_{u|\tilde{y},\boldsymbol{x}} = \Pr\left(U = 1 \mid \tilde{Y} = \tilde{y}, X = \boldsymbol{x}\right)$ | $\pi_{y,\boldsymbol{x}} = \Pr(Y = y, X = \boldsymbol{x})$ | Data-driven discovery tasks where $\tilde{Y}$ is an experimental outcome confirmed by a hypothesis test with type I/II error [16, 39]. |

**Table 1:** Common noise models that we consider in this work. We represent each model as a probability distribution with parameters $p_{u|y,\boldsymbol{x}}$ and show its corresponding probabilistic graphical model (PGM). Given a noisy dataset, noise model, and prior distribution $\pi_y$, we infer noise draws from a posterior distribution with parameters $q_{u|\tilde{y},\boldsymbol{x}}$.

## 3 REGRETFUL DECISIONS

Consider a practitioner who learns a model $f : \mathcal{X} \to \mathcal{Y}$ from a noisy dataset. In practice, they may learn a model that performs well on average. However, they cannot determine where it makes mistakes. In such tasks, individuals are subject to a *lottery of mistakes*. We characterize this effect in terms of *regret*.

**Definition 2.** Given a classification task where we learn a model $f : \mathcal{X} \to \mathcal{Y}$ from a noisy dataset, we define the *regret* for an instance $(\boldsymbol{x}_i, \tilde{y}_i)$ as:

$$\mathrm{Regret}(f(\boldsymbol{x}_i), \tilde{y}_i, U_i) := \mathbb{I}\left[e^{\mathrm{pred}}(f(\boldsymbol{x}_i), \tilde{y}_i) \neq e^{\mathrm{true}}(f(\boldsymbol{x}_i), y_i(U_i))\right]$$

Here:

- $e^{\mathrm{true}}(f(\boldsymbol{x}_i), y_i(U_i)) := \mathbb{I}[f(\boldsymbol{x}_i) \neq y_i(U_i)]$ indicates an *actual mistake* with respect to the true label. We write the true label as $y_i(U_i) := \tilde{y}_i \oplus U_i$ to show that it is a random variable.
- $e^{\mathrm{pred}}(f(\boldsymbol{x}_i), \tilde{y}_i)$ indicates the model has made an *anticipated mistake* – i.e., that it appears to have made a mistake based on what we can tell during training.

In practice, $e^{\mathrm{pred}}(\cdot)$ is determined by how we account for noise, if at all. If we ignore label noise and fit a model via standard ERM on the noisy dataset, then $e^{\mathrm{pred}}(f(\boldsymbol{x}_i), \tilde{y}_i) := \mathbb{I}[f(\boldsymbol{x}_i) \neq \tilde{y}_i]$. If we fit a model via noise-tolerant ERM [e.g., 40, 44], then $e^{\mathrm{pred}}(f(\boldsymbol{x}_i), \tilde{y}_i) := \tilde{\ell}_{01}(f(\boldsymbol{x}_i), \tilde{y}_i)$ where $\tilde{\ell}_{01}(\cdot)$ is an unbiased loss defined such that $\mathbb{E}_U[\tilde{\ell}_{01}(\boldsymbol{x}_i, \tilde{y}_i)] = \ell_{01}(f(\boldsymbol{x}_i), y_i)$.

Regret captures the irreducible error we incur due to randomness. In online learning, regret arises because we cannot foresee randomness in the future. In learning from noisy labels, regret arises because we cannot infer randomness from the past. In this case, randomness undermines our ability to determine which predictions are correct. In this regime, as individual predictions cannot be assumed to be accurate even on the training data. As a result, we cannot rely on predictions to support individual decisions. Moreover, we cannot rely on any downstream applications that depend on the correctness of individual predictions – e.g., model explanations [6, 48] or post-hoc analyses [25, 26, 34]. In Prop. 3, we explore the relationship between these effects and label noise.

**Proposition 3.** In a classification task where we learn a classifier $f$ from a noisy dataset $\tilde{\mathcal{D}}$:

$$\mathbb{E}_{U|X,\tilde{Y}} \left[ \text{Regret}(f(X), \tilde{Y}, U) \right] = \Pr(U = 1 \mid \tilde{Y}, X).$$

Prop. 3 provides an opportunity to highlight several implications of learning from label noise at the instance-level. On the one hand, this result implies that regret is *unavoidable* when learning under label noise. In practice, we can only avoid it by "predicting less" (e.g., via selective classification) or by "removing noise" (e.g., via relabeling). On the other hand, the result also implies that we can estimate the *expected* number of regretful predictions in terms of the posterior noise rate. In practice, however, we cannot tell *how* these mistakes are distributed over all instances.

One of the key issues in this regime is that the value of a prediction may be compromised, as each instance where $q_{u|\boldsymbol{x},\tilde{y}} > 0$ is subject to a lottery of mistakes. Consider screening for a rare disease using a diagnostic test. In such cases, we can view the presence of the disease as a clean label $y_i$ and the test outcome as a noisy label $\tilde{y}_i$. Given a disease that affects 10% of patients and a class-level noise model that flips 10% of positive cases, an average draw of label noise would affect 1% of predictions. In practice, such conditions would undermine the value of screening because any patient with a negative test may have the disease. We characterize these effects by measuring the proportion of instances in a dataset that are susceptible to regret – i.e., that are subject to the lottery of mistakes. Given a noisy dataset $\tilde{\mathcal{D}}$ and a posterior noise model $\Pr\left(U = 1 \mid X, \tilde{Y}\right)$, the number of points susceptible to regret is:

$$\text{Susceptibility}(\tilde{\mathcal{D}}) := \frac{1}{n} \sum_{i=1}^{n} \mathbb{I}\left[ \Pr\left(U = 1 \mid X = x_i, \tilde{Y} = \tilde{y}_i\right) > 0 \right] \tag{1}$$

**On the Regret of Hedging** One of the benefits of studying regret in this regime is that we can characterize when learning is feasible at both the population and instance-levels. Many algorithms for learning from noisy labels are designed to *hedge* against label noise [46]. Given a noisy dataset and a noise model, hedging minimizes the *expected risk over all possible noise draws*. In some cases, algorithms may implement this strategy explicitly via ERM with a modified loss [see e.g., 37, 40]. In others, algorithms may hedge implicitly – e.g., by assigning sample weights to training instances and setting their values to minimize expected risk over all possible draws [see e.g., 33, 44, 54].

In a best-case scenario, where we correctly specify the noise model and fit a model that minimizes the average number of mistakes over all noise draws, we would still incur regret. Formally, we would expect $\mathbb{E}_{U|X,Y}[\Delta\text{Error}(f, \tilde{\mathcal{D}}, U)] = 0$ where:

$$\Delta\text{Error}(f, \tilde{\mathcal{D}}; U) := \underbrace{\sum_{i=1}^{n} e^{\text{pred}}(f(\boldsymbol{x}_i), \tilde{y}_i)}_{\text{Predicted Training Error}} - \underbrace{\sum_{i=1}^{n} e^{\text{true}}(f(\boldsymbol{x}_i), y_i)}_{\text{True Training Error}} \tag{2}$$

However, the resulting model $f$ would still incur regret $\mathbb{E}_{U|X,\tilde{Y}} \left[ \text{Regret}(f, \tilde{\mathcal{D}}, U) \right] > 0$. In Prop. 4, we show that the classical hedging algorithm of Natarajan et al. [40] exhibits this behavior.

**Proposition 4.** Consider training a model $f : \mathcal{X} \to \mathcal{Y}$ on a noisy dataset via ERM with a modified loss function $\tilde{\ell} : \mathcal{Y} \times \mathcal{Y} \to \mathbb{R}_+$ such that $\mathbb{E}_U[\tilde{\ell}(f(\boldsymbol{x}), \tilde{y})] = \ell(f(\boldsymbol{x}), y)$ for all $(\boldsymbol{x}, \tilde{y})$. In this case, the model minimizes risk for an *implicit noise draw* $\boldsymbol{u}^{\text{mle}} = [u_1^{\text{mle}}, \ldots, u_n^{\text{mle}}]$ where $u_i^{\text{mle}}$ corresponds to most likely outcome under the posterior noise model $q_{u|\tilde{y}_i, \boldsymbol{x}_i}$.

Prop. 4 implies that hedging will incur regret unless the implicit noise draw $\boldsymbol{u}^{\text{mle}}$ matches the true noise in the dataset $\boldsymbol{u}^{\text{true}}$. In practice, this event is unlikely as $\lim_{n\to\infty} \Pr\left(\boldsymbol{u}^{\text{mle}} = \boldsymbol{u}^{\text{true}}\right) = 0$ (see Appendix A).

## 4 ANTICIPATING MISTAKES WITH PLAUSIBLE MODELS

Our results in Section 2 show how a model we learn under label noise will output regretful predictions. In this section, we develop methods to estimate the likelihood of an individual instance yielding a regretful prediction.

### 4.1 MOTIVATION

Our goal is to evaluate the correctness of individual predictions for models learned from noisy data. In a standard classification task, we apply an algorithm for ERM to a clean dataset, recover the model $\hat{f} \in \operatorname{argmin}_{f \in \mathcal{F}} \frac{1}{n} \sum_{i=1}^{n} \mathbb{I}[f(\boldsymbol{x}_i) \neq y_i]$, and evaluate the correctness of each prediction on the training data in terms of mistakes. When we learn from a noisy dataset $\tilde{D} = \{(\boldsymbol{x}_i, \tilde{y}_i)\}_{i=1}^{n}$, the corresponding measure is no longer deterministic:

$$\text{Mistake}(\boldsymbol{x}_i, Y_i, \hat{F}) = \mathbb{I}\left[\hat{F}(\boldsymbol{x}_i) \neq Y_i\right]. \tag{3}$$

In this case, the randomness stems from: (1) the true label $Y_i$, which is a random variable that can only be inferred from the observed noisy label $\tilde{y}_i$ and the posterior noise model; (2) the model $\hat{F} : \mathcal{X} \to \mathcal{Y}$, which is the output of a learning algorithm on the noisy dataset.

Our proposed measure, which we call *ambiguity*, quantifies the expected likelihood of a learning algorithm making a mistake on the training data – i.e., the expected value of (3).

$$\text{Ambiguity}(\boldsymbol{x}_i, \tilde{y}_i) := \mathbb{E}_{Y_i, \hat{F}|\tilde{D}}\left[\text{Mistake}(\boldsymbol{x}_i, Y_i, \hat{F})\right] = \mathbb{E}_{\boldsymbol{u} \sim U|\tilde{D}}\left[\mathbb{I}[\hat{F}(\boldsymbol{x}_i) \neq (\tilde{y}_i \oplus U_i)]\right] \tag{4}$$

Ambiguity uses all the information we have at hand: a noisy dataset and a noise model. In Prop. 5, we show how ambiguity corresponds to regret as it correctly ranks the instances based on the likelihood of experiencing regret.

**Proposition 5.** Given a classification task, denote the clean label risk of a model $\hat{F}(\boldsymbol{x}_i)$ on an instance $\boldsymbol{x}_i$ as $e$, that is $e := \Pr\left(\hat{F}(\boldsymbol{x}_i) \neq y_i\right)$. When $e < 0.5$ – that is, a model makes more correct than incorrect predictions – a higher label noise rate for instance $\boldsymbol{x}_i$ corresponds to higher Ambiguity$(\boldsymbol{x}_i, \tilde{y}_i)$.

Since Prop. 3 establishes that regret corresponds to the posterior noise rate, Prop. 5 suggests that ambiguity serves as a viable measure of regret, given its correspondence to the posterior noise rate.

### 4.2 ESTIMATION

We can construct unbiased estimates of ambiguity using Algorithm 1. Given a noisy dataset and a noise model, this procedure generates plausible realizations of a clean dataset, and then trains a set of plausible models that can be used to estimate ambiguity. In what follows, we describe this procedure in greater detail.

**Sampling Plausible Draws** Given a noisy dataset $\tilde{\mathcal{D}}$, class-level noise model $p_u$, and prior distribution $\pi_y := \Pr(Y = y)$, we can sample noise draws from the posterior distribution:

$$q_{u|\tilde{y}} = \frac{(1 - \pi_{\tilde{y}}) \cdot p_{u|1-\tilde{y}}}{p_{u|\tilde{y}} \cdot (1 - \pi_{\tilde{y}}) + (1 - p_{u|\tilde{y}}) \cdot \pi_{\tilde{y}}} \tag{5}$$

This generalizes to different types of noise models (see e.g., Table 1). We can use these samples from the posterior distribution to estimate ambiguity directly. In practice, however, it may lead to biased estimates by returning *atypical draws* – unlikely noise draws under a given noise model (e.g., a noise draw that flips 30% of labels under a uniform noise model with a noise rate of 10%). In settings where we wish to estimate ambiguity using a limited number of draws, an atypical draw can bias our estimates and undermine their utility. Although we could moderate this bias by increasing the number of draws, this would require training a separate model for each draw. We address these issues by sampling from a set of plausible draws.

---

**Algorithm 1** Generate Plausible Draws, Datasets, and Models

---

**Input** noisy dataset $(\boldsymbol{x}_i, \tilde{y}_i)_{i=1}^n$, noise model $p_{u|y}$, number of models $m \geq 1$, atypicality $\epsilon \in [0, 1]]$
**Initialize** $\hat{\mathcal{F}}_\epsilon^{\text{plaus}} \leftarrow \{\}$
1: **repeat**
2:     $u_i \sim \text{Bernouilli}(q_{u|\tilde{y}, \boldsymbol{x}})$ for $i \in [n]$                             *generate noise draw by posterior inference*
3:     **if** $[u_1, \dots, u_n] \in \mathcal{U}_\epsilon$ **then**                             *check if draw is plausible using Def. 6*
4:         $\hat{y}_i \leftarrow \tilde{y}_i \oplus u_i$ for $i \in [n]$
5:         $\hat{\mathcal{D}} \leftarrow \{(\boldsymbol{x}_i, \hat{y}_i)\}_{i=1}^n$                              *construct plausible clean dataset*
6:         $\hat{f} \leftarrow \operatorname{argmin}_{f \in \mathcal{F}} \hat{R}(f; \hat{\mathcal{D}})$                         *train plausible model*
7:         $\hat{\mathcal{F}}_\epsilon^{\text{plaus}} \leftarrow \hat{\mathcal{F}}_\epsilon^{\text{plaus}} \cup \{\hat{f}\}$                       *update plausible models*
8:     **end if**
9: **until** $|\hat{\mathcal{F}}_\epsilon^{\text{plaus}}| = m$
**Output** $\hat{\mathcal{F}}_\epsilon^{\text{plaus}}$, sample of $m$ models from the set of plausible models $\mathcal{F}_\epsilon^{\text{plaus}}$

---

**Definition 6.** Given a noise draw $\boldsymbol{u} \in \{0, 1\}^n$, denote its true posterior noise rate as $q_{u|\tilde{y}} := \Pr(U = 1 \mid \tilde{Y} = \tilde{y})$ and empirical noise rate as $\hat{q}_{u|\tilde{y}} := \frac{1}{n}\sum_{i=1}^n \mathbb{I}[u_i = 1 \mid \tilde{y}_i = y]$. For any $\epsilon \in [0, 1]$, the *set of plausible draws* contains all draws whose empirical noise rate is within $\epsilon$ of the posterior rate:

$$\mathcal{U}_\epsilon(\tilde{\boldsymbol{y}}) := \{\boldsymbol{u} \in \{0, 1\}^n \text{ s.t. } |q_{u|\tilde{y}} - \hat{q}_{u|\tilde{y}}| < \epsilon \cdot q_{u|\tilde{y}} \text{ for all } u \in \{0, 1\}\}.$$

The set of plausible draws is a strongly typical set [see 9]. In a classification task where $n$ is large, we can expect most (but not all) draws to concentrate in $\mathcal{U}_\epsilon$ [see Theorem 3.1.2 in 9]. We can limit atypical draws by setting the *atypicality parameter* $\epsilon$, which represents the relative deviation between the true noise rate $q_{u|\tilde{y}}$ and the noise rate of sampled draws. Given a uniform noise model where $q_{u|\tilde{y}} = 0.1$, we would set $\epsilon = 0.2$ to only consider draws that flip between 8% to 12% of instances. Alternatively, we can set $\epsilon$ to ensure that $\mathcal{U}_\epsilon(\tilde{\boldsymbol{y}})$ will include a particular noise draw $\boldsymbol{u}^* \in \mathcal{F}_\epsilon^{\text{plaus}}$ with high probability (see Prop. 9 in Appendix A). By default, we set $\epsilon = 0.1$ to consider draws within 10% of what we would expect.

**Estimating Ambiguity**     Given a plausible noise draw $\boldsymbol{u}^k \in \mathcal{U}_\epsilon(\tilde{\boldsymbol{y}})$, we construct a *plausible* clean dataset by pairing each $\boldsymbol{x}_i$ with a *plausible* value of true label $\hat{y}_i^k = u^k \oplus \tilde{y}_i$.

**Definition 7.** The *set of $\epsilon$-plausible models* contains all models trained using $\epsilon$-plausible datasets:

$$\mathcal{F}_\epsilon^{\text{plaus}} := \left\{\hat{f} \in \operatorname*{argmin}_{f \in \mathcal{F}} \hat{R}(f, \hat{\mathcal{D}}) \mid \hat{\mathcal{D}} := \{(\boldsymbol{x}_i, \hat{y}_i^k)\}_{i=1}^n, \boldsymbol{u} \in \mathcal{U}_\epsilon(\tilde{\boldsymbol{y}})\right\}.$$

In an ideal case, where we recover a plausible draw that matches the true draw $\boldsymbol{u}^k = \boldsymbol{u}^{\text{true}}$, our procedure returns a plausible dataset $\hat{\mathcal{D}}^k$ and model $\hat{f}^k$ that perfectly flags all regretful predictions. Seeing how $\boldsymbol{u}^{\text{true}}$ is unknown, we repeat this process $m$ times and use the $m$ plausible models to get an *unbiased* estimate of ambiguity for each point in our noisy dataset as:
$\hat{\mu}(\boldsymbol{x}, \tilde{y}) := \frac{1}{m}\sum_{k \in [m]} \mathbb{I}\left[\hat{f}^k(\boldsymbol{x}) \neq \hat{y}^k\right]$.

In practice, we can use ambiguity as a confidence score to operationalize techniques to learn or predict reliably. We propose a few examples and demonstrate how these perform in Section 6:

- *Data Cleaning*: We can use ambiguity to flag regretful instances in a training dataset to drop or relabel. Given the correspondence between regret and noise (Prop. 3), this approach can be used to "de-noise" a dataset to train models that generalize better on clean test data.

- *Selective Prediction*: We can use ambiguity to abstain from potentially regretful predictions at test time via selective prediction [12]. This approach can be used in instances such as clinical decision support, where we only show sufficiently reliable predictions and defer uncertain predictions to a clinician.

**Discussion**     The main limitation of this approach is that we assume access to a correctly specified noise model. This assumption is a practical limitation and can be validated by comparing it against

distributions estimated from a noisy dataset [see e.g., 31, 33, 44]. When working with simple noise models (e.g., uniform or class-level), we may be conservative and assume a higher noise rate. Alternatively, we can hedge against misspecification by setting $\epsilon$ to capture a larger set of plausible draws. The set of plausible models can also be used in ways to construct uncertainty measures, as demonstrated in Sections 5 and 6.

We also note that our estimates of ambiguity assume that the true noise draw, $\boldsymbol{u}^{\text{true}}$, is typical. In practice, although $\boldsymbol{u}^{\text{true}}$ is unknown, most draws can be shown to be typical – this follows from a standard application of a Chernoff bound [9].

## 5 EXPERIMENTS

In this section, we present an empirical study on clinical prediction tasks. Our goals are to document the effects of label noise on individual-level predictions. Supporting material and code can be found in Appendix B and GitHub.

**Setup** We work with 5 classification datasets from clinical applications where models support individual medical decisions (see Table 3). We treat the labels in each dataset as true labels. We create noisy datasets by corrupting the labels using a noise draw sampled according to three class-level noise models with noise rates $[5\%, 20\%, 40\%]$ where label noise only affects positive instances ($y_i = 1$). We split each dataset into a training sample (80%), which we use to train a logistic regression model (LR) and a neural network (DNN) using noisy labels, and a test sample (20%), which we use to measure out-of-sample performance using true labels. We train these models using the following methods:

1. Ignore, where we ignore label noise and fit a model to predict noisy training labels; and
2. Hedge where we hedge against label noise using the method of Natarajan et al. [40].

This yields 12 models for each dataset (3 noise regimes $\times$ 2 model classes $\times$ 2 training procedures).

| Metric | Definition | Description |
|---|---|---|
| TrueError$(f, \tilde{\mathcal{D}})$ | $\frac{1}{n} \sum_{i \in [n]} e^{\text{true}}(f(\boldsymbol{x}_i), y_i)$ | Error rate of $f$ on the *clean* training labels. |
| AnticipatedError$(f, \tilde{\mathcal{D}})$ | $\frac{1}{n} \sum_{i \in [n]} e^{\text{pred}}(f(\boldsymbol{x}_i), \tilde{y}_i) - e^{\text{true}}(f(\boldsymbol{x}_i), y_i)$ | Error rate of $f$ on the *noisy* labels. |
| Susceptibility$(\tilde{\mathcal{D}})$ | $\frac{1}{n} \sum_{i \in [n]} \mathbb{I}\left[\Pr\left(U = 1 \mid X = x_i, \tilde{Y} = \tilde{y}_i\right) > 0\right]$ | Proportion of instances in $\tilde{\mathcal{D}}$ subject to regret. |
| Regret$(f, \tilde{\mathcal{D}})$ | $\frac{1}{n} \sum_{i \in [n]} \mathbb{I}\left[e^{\text{pred}}(f(\boldsymbol{x}_i), \tilde{y}_i) \neq e^{\text{true}}(f(\boldsymbol{x}_i), y_i)\right]$ | Mean regret across all instances in $\tilde{\mathcal{D}}$. We expect Regret$(f, \tilde{\mathcal{D}}) \approx \sum_y q_{u\mid y} \cdot \pi_y$ under class-level label noise. |
| Overreliance$(f, \tilde{\mathcal{D}})$ | $\frac{1}{n} \sum_{i \in [n]} \mathbb{I}\left[e^{\text{true}}(f(\boldsymbol{x}_i), y_i) = 1, e^{\text{pred}}(f(\boldsymbol{x}_i), \tilde{y}_i) = 0\right]$ | Proportion of predictions in $\tilde{\mathcal{D}}$ that are incorrectly perceived as accurate. |

**Table 2:** Overview of summary statistics in Table 3. We report these metrics for models that we train from noisy labels using a specific training procedure, model class, noise model, and dataset. We evaluate all models trained on a given dataset and noise model using a fixed noise draw.

We characterize the accuracy and reliability of predictions from each model using the measures in Table 2. We report our results for LR models in Table 3 and results for DNN models in Appendix B. In what follows, we discuss all results.

**On Label Noise, Regret, and Hedging** Our results in Table 3 highlight several implications of learning under label noise. We confirm that our result in Prop. 3 holds empirically – i.e., the expected prevalence of regretful predictions corresponds to the effective noise rate in each dataset. We observe similar effects across all datasets, model classes, and noise regimes, underscoring the need to quantify the effect of label noise on individual predictions.

Existing approaches to handle label noise (i.e., hedging) can learn models that are robust to noise at a population-level but still experience regret. As shown in Table 3, we observe that Hedge can moderate the impact of label noise at a population-level by reducing the true (clean label) error compared to Ignore. Even with more noise robustness, regret is unchanged, and remains high across

all experimental conditions. On the `mortality` dataset, for example, Hedge reduces the error rate by over 13% compared to Ignore for a LR model under 40% label noise. However, regret is unchanged and continues to affect $19.5\%$ of instances. It is interesting to note that Hedge can moderate the effects of overrelianceby *redistributing* unforeseen mistakes from instances that lead to overreliance to instances where $e^{\text{pred}}(f(\boldsymbol{x}_i), \tilde{y}_i) = 1$ and $e^{\text{true}}(f(\boldsymbol{x}_i), y_i) = 0$ – i.e., where a practitioner may fail to reap the benefits of a correct prediction because it appears to be incorrect.

**On the Lottery of Mistakes** Our results highlight how a small amount of label noise can undermine common use cases for prediction by subjecting a far greater number of instances to a lottery of mistakes. In Table 3, for example, we consider a noise model where only positive instances ($y = 1$) are subject to label noise. Thus, every instance with a negative noisy label ($\tilde{y} = 0$) is subject to a lottery of mistakes. We report the proportion of points that take part in the lottery using the susceptibility metric in Eq. (1). In this case, we can see that in a task where the label noise is as low as 5%, over half of instances are subject to lottery across all five datasets. For example, in the `lungcancer` dataset, even a small misdiagnosis (i.e., label noise) rate, which is inevitable, can compromise the reliability of half of all diagnoses.

| Dataset | Metrics | $p_{u\|y=1} = 5\%$ | | $p_{u\|y=1} = 20\%$ | | $p_{u\|y=1} = 40\%$ | |
|---|---|---|---|---|---|---|---|
| | | Ignore | Hedge | Ignore | Hedge | Ignore | Hedge |
| `shock_eicu` $n = 3,456$ $d = 104$ Pollard et al. [45] | True Error | 24.4% | 23.5% | 27.1% | 24.6% | 41.0% | 24.3% |
| | Anticipated Error | 25.7% | 25.2% | 28.3% | 29.4% | 28.2% | 33.5% |
| | Regret | 3.0% | 3.0% | 10.1% | 10.1% | 19.7% | 19.7% |
| | Overreliance | 1.1% | 0.9% | 6.3% | 3.8% | 22.6% | 7.9% |
| | Susceptibility | 52.6% | 52.6% | 59.7% | 59.7% | 69.3% | 69.3% |
| `shock_mimic` $n = 15,254$ $d = 104$ Johnson et al. [22] | True Error | 20.8% | 20.2% | 25.0% | 20.3% | 34.9% | 20.1% |
| | Anticipated Error | 22.1% | 21.7% | 26.8% | 26.4% | 27.4% | 32.5% |
| | Regret | 2.5% | 2.5% | 10.2% | 10.2% | 19.8% | 19.8% |
| | Overreliance | 0.8% | 0.6% | 5.8% | 2.8% | 18.8% | 5.5% |
| | Susceptibility | 52.5% | 52.5% | 60.2% | 60.2% | 69.8% | 69.8% |
| `lungcancer` $n = 62,916$ $d = 40$ NCI [41] | True Error | 31.7% | 30.8% | 33.7% | 30.8% | 43.0% | 31.1% |
| | Anticipated Error | 32.2% | 31.5% | 32.7% | 33.6% | 30.0% | 36.5% |
| | Regret | 2.5% | 2.5% | 10.0% | 10.0% | 19.7% | 19.7% |
| | Overreliance | 1.5% | 1.3% | 8.1% | 5.4% | 23.4% | 11.3% |
| | Susceptibility | 52.7% | 52.7% | 60.2% | 60.2% | 69.9% | 69.9% |
| `mortality` $n = 20,334$ $d = 84$ Le Gall et al. [28] | True Error | 19.5% | 19.0% | 23.2% | 19.1% | 33.2% | 19.4% |
| | Anticipated Error | 20.7% | 20.4% | 25.7% | 25.0% | 27.7% | 30.9% |
| | Regret | 2.2% | 2.2% | 9.8% | 9.8% | 19.5% | 19.5% |
| | Overreliance | 0.6% | 0.5% | 4.9% | 2.6% | 17.3% | 5.8% |
| | Susceptibility | 52.2% | 52.2% | 59.8% | 59.8% | 69.5% | 69.5% |
| `support` $n = 9,696$ $d = 114$ Knaus et al. [24] | True Error | 33.1% | 33.7% | 36.7% | 33.7% | 44.2% | 33.9% |
| | Anticipated Error | 33.4% | 34.1% | 34.1% | 36.0% | 29.7% | 38.6% |
| | Regret | 2.6% | 2.6% | 10.0% | 10.0% | 19.6% | 19.6% |
| | Overreliance | 1.8% | 1.7% | 9.6% | 6.0% | 24.3% | 12.1% |
| | Susceptibility | 52.6% | 52.6% | 60.0% | 60.0% | 69.6% | 69.6% |

**Table 3:** Accuracy and reliability of predictions for LR models trained on noisy datasets where we flip 5%, 20% and 40% of positive instances. We defer results for DNN models to Appendix B for clarity.

**On the Consequences of Blindness** Our results highlight the importance of considering the effect of label noise in instance-level predictions. Many real-world practitioners assume that their training data is clean and ignore label noise, but most real-world datasets are not perfectly labeled – this inevitably leads to regretful predictions on individuals.

To demonstrate how regretful predictions can negatively impact individuals, we consider a particular flavor of regret – *overreliance* – the fraction of instances where a practitioner would incorrectly assume that a model assigned a correct prediction – i.e. where $e^{\text{pred}}(f(\boldsymbol{x}_i), \tilde{y}_i) = 0$ and $e^{\text{true}}(f(\boldsymbol{x}_i), y_i) = 1$. From Table 3, we consider overreliance on the `lungcancer` dataset, under $40\%$ noise. We observe that up to $23.4\%$ of instances are assigned this type of prediction. In practice, such instances correspond to patients with cancer but who are classified as cancer-free based on the prediction of a seemingly accurate model. These are patients where the model is making a mistake, however, the practitioner cannot tell as it would not appear to be a mistake based on the noisy label. This highlights the importance of looking at the distribution of regretful instances across predictions. By analyzing the distribution of regretful predictions, we can adjust our reliance on model predictions – ensuring that practitioners do not blindly trust or explain away incorrect model decisions.

## 6 DEMONSTRATIONS

In this section, we show how the machinery developed in Section 4 can be used to promote safety at critical parts of the machine learning lifecycle in real-world applications where noisy labels are inevitable.

**Data Cleaning** Our approach in Section 4 can *clean* noisy datasets by using ambiguity to drop noisy instances from a training dataset. Given the "denoised" dataset, we can then train models that perform better in deployment. In Fig. 2, we demonstrate the effectiveness of this approach on the `shock_mimic` dataset. Here, we drop training examples using a confidence-based threshold rule of the form $\mathbb{I}[\text{conf}(\boldsymbol{x}_i) \leq \tau]$, where $\tau$ is a threshold set to control the number of instances to drop. We

compare the performance of this strategy using confidence scores that we can compute on training data: (1) $\text{conf}(\boldsymbol{x}_i) = 1 - \hat{\mu}(\boldsymbol{x}_i, \tilde{y}_i)$, which is a measure based on the estimated ambiguity that we recover using Algorithm 1; and (2) $\text{conf}(\boldsymbol{x}_i) = \hat{p}(y_i \mid \boldsymbol{x}_i)$, which is the predicted probability of a final model. As shown, removing instances with high ambiguity from the training dataset prior to training a final model on the cleaned data leads to improved test error on clean labels. Specifically, using ambiguity to drop uncertain instances reduces test error by 14.9% when dropping only 20% of noisy training data compared to a baseline approach.

**Selective Classification with Cheap Labels** We use our results to highlight how the machinery in Section 4 can promote safer predictions. Consider the `shock_mimic` dataset in Fig. 3 – here, we use the same confidence-based threshold rule of the form $\mathbb{I}\left[\text{conf}(\boldsymbol{x}_i) \leq \tau\right]$ where $\text{conf}(\boldsymbol{x}_i)$ is a confidence score and $\tau$ is a threshold value. We consider confidence scores based on standard predicted probabilities and ambiguity, where ambiguity can be measured using cheaply acquired test instances (e.g., noisy test data). We show how the selective test error on clean labels and selective regret change as we vary the confidence threshold value $\tau \in (0, 1)$. Specifically, in a regime where 20% of the labels are noisy, abstaining on 40% of instances using ambiguity reduces selective error by -6.6% and selective regret by -5.9% compared to the standard approach on `cshock_mimic`.

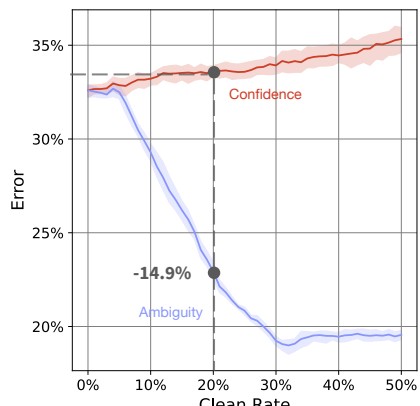

**Figure 2:** Clean test error for a LR model on the `shock_mimic` dataset with 40% class-level label noise when dropping training instances using different confidence-based threshold rules. We show the clean test error vs percent of instances dropped from training using confidence measures based on predicted probabilities $\text{conf}(\boldsymbol{x}_i) := \hat{p}(\tilde{y}_i \mid \boldsymbol{x}_i)$ or ambiguity $\text{conf}(\boldsymbol{x}_i) := 1 - \hat{\mu}(\boldsymbol{x}_i, \tilde{y}_i)$.

**Selective Classification for Scientific Discovery** We demonstrate how our approach can support a modern scientific discovery task in biotechnology. In such tasks, researchers perform in-vitro experiments to identify instances with desired properties [e.g., identifying new antibiotics 50]. Given a dataset of successful and unsuccessful experiments and their characteristics, we can train a model to predict which future experiments are likely to succeed, thereby accelerating discovery by prioritizing high-yield experiments.

We use the `enhancer` dataset from Gschwind et al. [16] to predict the outcome of experiments to discover *enhancers* – i.e., segments of DNA that regulate gene expression. The dataset contains $n = 992$ noisy instances $(\boldsymbol{x}_i, \tilde{y}_i)$, each with $d = 13$ features (e.g., gene location, cell type, etc.).

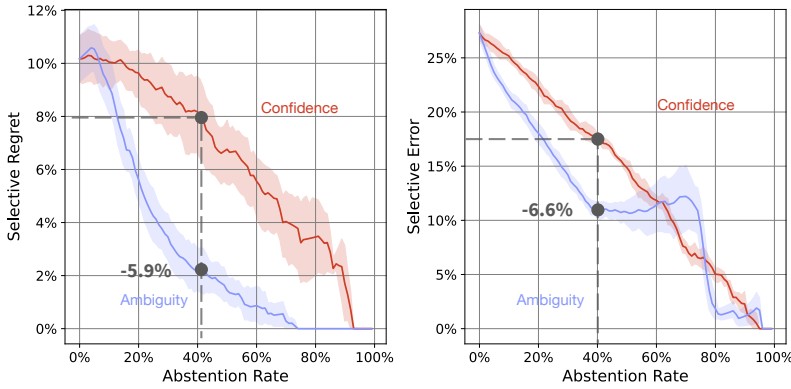

**Figure 3:** Selective classification frontiers for a LR model on the `shock_mimic` dataset under 20% class-level noise when abstaining from uncertain predictions at test time using a confidence-based threshold rule. We plot the selective regret (left) and selective error (right) as we vary the percent of abstained predictions for confidence measures based on predicted probabilities $\text{conf}(\boldsymbol{x}_i) := \hat{p}(\tilde{y}_i \mid \boldsymbol{x}_i)$ and ambiguity $\text{conf}(\boldsymbol{x}_i) := 1 - \hat{\mu}(\boldsymbol{x}_i)$.

In this setting, a noisy label $\tilde{y}_i = 1$ indicates a statistically significant experiment (i.e., reject null hypothesis: $H_0$ = "no effect"). Here, label noise arises from the Type I error for each experiment:

$$\Pr\left(\tilde{y}_i = 1 \mid y_i = 0\right) = \Pr\left(\text{reject } H_0 \mid H_0 \text{ holds}\right) = p\text{-value for experiment } i$$

We fit a LR model $f$ and use its predictions to identify experiments that are likely to succeed in the test sample. To avoid low-confidence predictions, we use a thresholding rule of the form $\mathbb{I}\left[\text{conf}(\boldsymbol{x}_i) \leq \tau\right]$ where $\tau$ is a threshold value that we can set to control the number of experiments to perform. Confidence for an instance is measured with a score $\text{conf}(\boldsymbol{x}_i) := 1 - \text{Disagreement}(\boldsymbol{x}_i)$ where $\text{Disagreement}(\boldsymbol{x}_i)$ measures the *disagreement* between the predictions of $f$ and the $m$ plausible models from Algorithm 1:

$$\text{Disagreement}(\boldsymbol{x}_i) := \frac{1}{m} \sum_{k \in [m]} \mathbb{I}\left[\hat{f}^k(\boldsymbol{x}_i) \neq f(\boldsymbol{x}_i)\right] \tag{6}$$

Fig. 4 shows that disagreement can reliably predict which experiments will be successful. We use two strategies for abstention: (1) a standard approach thresholding according to $\hat{p}(\tilde{y}_i \mid \boldsymbol{x}_i)$, or (2) using disagreement rates (6). Performance is measured using test *hit rate* (i.e., the number of successful experiments divided by the number of total experiments that we run). Our approach improves the hit rate (+10.7%) compared to standard confidence-based abstention, with a modest 20% abstention rate (Fig. 4). This demonstrates that we can optimize laboratory resource allocation and increase the discovery rate of enhancers by forgoing 20% of experiments.

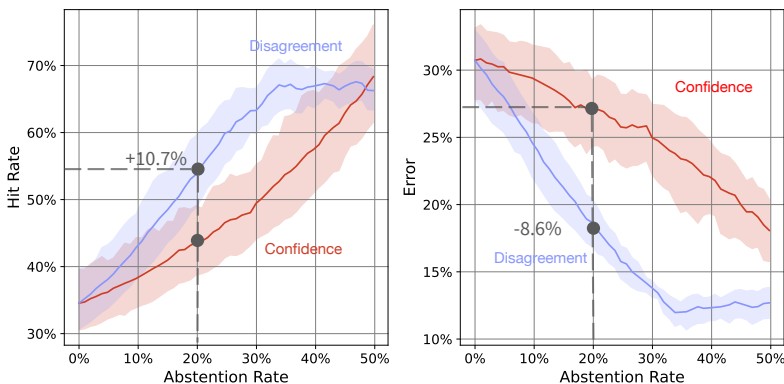

**Figure 4:** Selective classification frontiers for an LR model on the `enhancer` dataset when abstaining from uncertain predictions using a confidence-based threshold rule. We plot the selective hit rate (left) and selective error (right) as we increase the proportion of abstained predictions for confidence measures based on predicted probabilities $\text{conf}(\boldsymbol{x}_i) := \hat{p}(\tilde{y}_i \mid \boldsymbol{x}_i)$ and disagreement $\text{conf}(\boldsymbol{x}_i) := 1 - \text{Disagreement}(\boldsymbol{x}_i)$.

## 7 CONCLUDING REMARKS

Learning under label noise is a major challenge in practice. While models may perform well on average, a model that is 99% accurate can inadvertently misclassify *anyone*, as label noise can subject each individual prediction to a lottery of mistakes. In this work, we studied these effects through the lens of regret and highlighted the inherent limits of learning in this regime.

Our results show that, even as regret is inevitable when learning from label noise, we can operationalize simple techniques to predict safely by quantifying the uncertainty in individual predictions – e.g., by estimating ambiguity of individual predictions and using this to flag predictions where we should abstain or examples that we should re-label.

Our use of regret extends beyond label noise – into any setting where models are trained on datasets with a single draw of noise [e.g., for probabilistic classification 13]. In such settings, regret can explicitly reveal the impact of predictions on individuals, and estimating it can act as a safeguard or signal to collect more data or avoid prediction altogether.

ACKNOWLEDGMENTS

This work was supported by funding from the National Science Foundation IIS 2040880, IIS 2313105, IIS-2007951, IIS-2143895, and the NIH Bridge2AI Center Grant U54HG012510. Sujay Nagaraj was supported by a CIHR Vanier Scholarship. Resources used in preparing this research were provided, in part, by the Province of Ontario, the Government of Canada through CIFAR, and companies sponsoring the Vector Institute.

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

# A  OMITTED PROOFS

## A.1  RESULTS FROM SECTION 3

*Proof of Prop. 3.* Consider any classification task with label noise. Given a point with $(X, \tilde{Y})$, let $\rho_{X,\tilde{Y}} := \Pr\left(U = 1 \mid X, \tilde{Y}\right)$ denote the posterior noise rate and $\ell_{01}(f(X), \tilde{Y}) := \mathbb{I}\left[f(X) \neq \tilde{Y}\right]$ its zero-one loss.

By definition, a hedging algorithm [see e.g., 40] is designed to learn a classifier $f$ such that:

$$\mathbb{E}_{U|X,Y}[e^{\text{pred}}(f(X), \tilde{Y})] = e^{\text{true}}(f(X), Y).$$

We observe that $f$ will achieve zero error in expectation as a result of the unbiasedness property of hedging algorithms.

$$\mathbb{E}_{X,Y,U}\left[e^{\text{pred}}(f(X), \tilde{Y}) - e^{\text{true}}(f(X), Y)\right] = \mathbb{E}_{X,Y}E_{U|X,Y}\left[e^{\text{pred}}(f(X), \tilde{Y}) - e^{\text{true}}(f(X), Y)\right]$$
$$= 0$$

The last line follows from the fact that $\tilde{Y}$ is a deterministic function of $U$ given $Y$.

We now show that will still incur regret in this regime. We begin by expressing the expected regret for any point $(X, \tilde{Y})$ and any noise draw $U$ as:

$$\mathbb{E}_{X,\tilde{Y},U}\left[\text{Regret}(X, \tilde{Y}, U)\right]$$
$$= \mathbb{E}_{X,\tilde{Y}}\left[(1 - 2q_u) \cdot (e^{\text{pred}}(f(X), \tilde{Y}) + \ell_{01}(f(X), \tilde{Y})) + 2(q_u - 1) \cdot e^{\text{pred}}(f(X), \tilde{Y}) \cdot \ell_{01}(f(X), \tilde{Y}) + q_u\right]$$

$$\mathbb{E}_{X,\tilde{Y},U}[\text{Regret}(X, \tilde{Y}, U)] = \mathbb{E}_{X,\tilde{Y},U}\left[\mathbb{I}\left[e^{\text{pred}}(f(X), \tilde{Y}) \neq \mathbb{I}\left[f(X) \neq \tilde{Y}(1 - U) + (1 - \tilde{Y})U\right]\right]\right]$$
$$= \mathbb{E}_{X,\tilde{Y}}\mathbb{E}_{U|X,\tilde{Y}}\left[\mathbb{I}\left[e^{\text{pred}}(f(X), \tilde{Y}) \neq \mathbb{I}\left[f(X) \neq \tilde{Y}(1 - U) + (1 - \tilde{Y})U\right]\right]\right]$$
$$= \mathbb{E}_{X,\tilde{Y}}\mathbb{E}_{U|X,\tilde{Y}}\left[e^{\text{pred}}(f(X), \tilde{Y})(1 - \mathbb{I}\left[f(X) \neq \tilde{Y}(1 - U) + (1 - \tilde{Y})U\right])\right.$$
$$\left. + (1 - e^{\text{pred}}(f(X), \tilde{Y}))\mathbb{I}\left[f(X) \neq \tilde{Y}(1 - U) + (1 - \tilde{Y})U\right]\right]$$
$$= \mathbb{E}_{X,\tilde{Y}}\mathbb{E}_{U|X,\tilde{Y}}\left[e^{\text{pred}}(f(X), \tilde{Y})(1 - \mathbb{I}\left[f(X) \neq \tilde{Y}\right](1 - U) - \mathbb{I}\left[f(X) \neq 1 - \tilde{Y}\right]U)\right.$$
$$\left. + (1 - e^{\text{pred}}(f(X), \tilde{Y}))(\mathbb{I}\left[f(X) \neq \tilde{Y}\right](1 - U) + \mathbb{I}\left[f(X) \neq 1 - \tilde{Y}\right]U)\right]$$

Letting $q_u = \Pr\left(U = 1 \mid X, \tilde{Y}\right)$ and $\ell_{01}(f(X), \tilde{Y}) = \mathbb{I}\left[f(X) \neq \tilde{Y}\right]$, we have:

$$= \mathbb{E}_{X,\tilde{Y}}\left[(1 - q_u)(e^{\text{pred}}(f(X), \tilde{Y})(1 - \ell_{01}(f(X), \tilde{Y})) + (1 - e^{\text{pred}}(f(X), \tilde{Y}))\ell_{01}(f(X), \tilde{Y}))\right.$$
$$\left. + q_u(e^{\text{pred}}(f(X), \tilde{Y})(1 - \ell_{01}(f(X), 1 - \tilde{Y})) + (1 - e^{\text{pred}}(f(X), \tilde{Y}))\ell_{01}(f(X), 1 - \tilde{Y}))\right]$$
$$\mathbb{E}_{X,\tilde{Y},U}[\text{Regret}(X, \tilde{Y}, U)] = \mathbb{E}_{X,\tilde{Y}}\left[(1 - 2q_u) \cdot (e^{\text{pred}}(f(X), \tilde{Y}) + \ell_{01}(f(X), \tilde{Y}))\right.$$
$$\left. + 2(q_u - 1) \cdot e^{\text{pred}}(f(X), \tilde{Y}) \cdot \ell_{01}(f(X), \tilde{Y}) + q_u\right].$$

When there is no label noise, we have that $q_u = 0$ and $e^{\text{pred}}(f(X), \tilde{Y}) = \ell_{01}(f(X), \tilde{Y})$ for all $X, \tilde{Y}$. Because they are binary terms, in this regime, we have:

$$\mathbb{E}_{X,\tilde{Y},U}\left[\text{Regret}(X, \tilde{Y}, U)\right] = \mathbb{E}_{X,\tilde{Y}}[0] = 0$$

When there is label noise, we have that $q_u > 0$ for some $X, \tilde{Y}$. In this regime, we have:

$$\mathbb{E}_{X,\tilde{Y},U}\left[\text{Regret}(X, \tilde{Y}, U)\right] = \mathbb{E}_{X,\tilde{Y}}[q_u] > 0.$$

□

The proof for Prop. 4 uses the following lemma.

**Lemma 8.** Minimizing the expected risk under the clean label distribution is equivalent to minimizing a noise-corrected (hedged) risk under the noisy label distribution.

$$\mathbb{E}_{X,Y}\left[\mathbb{I}\left[f(X) \neq Y\right]\right] = \mathbb{E}_{X,\tilde{Y}}\left[(1 - q_u\mathbb{I}\left[f(X) \neq \tilde{Y}\right] + q_u\mathbb{I}\left[f(X) \neq 1 - \tilde{Y}\right]\right] \tag{7}$$

Here:

- $q_u = \frac{(1-\pi_{\tilde{y},\boldsymbol{x}})\cdot p_{u|1-\tilde{y},\boldsymbol{x}}}{p_{u|\tilde{y},\boldsymbol{x}}\cdot(1-\pi_{\tilde{y},\boldsymbol{x}})+(1-p_{u|\tilde{y},\boldsymbol{x}})\cdot\pi_{\tilde{y},\boldsymbol{x}}}$
- $\pi_{\tilde{y},\boldsymbol{x}} = \Pr\left(Y = \tilde{y}|X = \boldsymbol{x}\right)$ is the clean class prior an observed noisy label,
- $p_u = \Pr\left(U = 1 \mid Y = y, X = \boldsymbol{x}\right)$ is the class-level noise probability.

The result is analogous to Lemma 1 in Natarajan et al. [40]. In what follows, we include an additional proof for the sake of completeness.

*Proof.*

$$\text{ExpectedRisk}(f) = \mathbb{E}_{X,Y}\left[\mathbb{I}\left[f(X) \neq Y\right]\right]$$

$$= \mathbb{E}_{X,\tilde{Y},U}\left[\mathbb{I}\left[f(X) \neq \tilde{Y}(1-U) + U(1-\tilde{Y})\right]\right]$$

$$= \mathbb{E}_{X,\tilde{Y}}\mathbb{E}_{U|X,\tilde{Y}}\left[\mathbb{I}\left[f(X) \neq \tilde{Y}(1-U) + U(1-\tilde{Y})\right]\right]$$

$$= \mathbb{E}_{X,\tilde{Y}}\mathbb{E}_{U|X,\tilde{Y}}\left[\mathbb{I}\left[f(X) \neq \tilde{Y}\right](1-U) + \mathbb{I}\left[f(X) \neq 1 - \tilde{Y}\right]U\right]$$

$$= \mathbb{E}_{X,\tilde{Y}}\left[\mathbb{E}_{U|X,\tilde{Y}}[\mathbb{I}\left[f(X) \neq \tilde{Y}\right](1-U)] + \mathbb{E}_{U|X,\tilde{Y}}[\mathbb{I}\left[f(X) \neq 1 - \tilde{Y}\right]U]\right]$$

$$= \mathbb{E}_{X,\tilde{Y}}\left[\Pr\left(U = 0|\tilde{Y},X\right)\mathbb{I}\left[f(X) \neq \tilde{Y}\right] + \Pr\left(U = 1|\tilde{Y},X\right)\mathbb{I}\left[f(X) \neq 1 - \tilde{Y}\right]\right]$$

$$= \mathbb{E}_{X,\tilde{Y}}\left[\Pr\left(Y = \tilde{Y}|\tilde{Y},X\right)\mathbb{I}\left[f(X) \neq \tilde{Y}\right] + \Pr\left(Y \neq \tilde{Y}|\tilde{Y},X\right)\mathbb{I}\left[f(X) \neq 1 - \tilde{Y}\right]\right]$$

$$= \mathbb{E}_{X,\tilde{Y}}\left[(1 - q_u\mathbb{I}\left[f(X) \neq \tilde{Y}\right] + q_u\mathbb{I}\left[f(X) \neq 1 - \tilde{Y}\right]\right]$$

We can recover the statement of Lemma 8 by applying Bayes theorem to write $q_u$ in terms of the clean class priors and class-level noise probabilities. □

*Proof of Prop. 4.* We define $u^{\text{mle}}$ as the noise draw for instance $(X, Y)$, such that using $u^{\text{mle}}$ to minimize the expected risk implicitly coincides with the true minimizer of the expected risk (defined in Lemma 8). That is:

$$\operatorname*{argmin}_{f\in\mathcal{F}}\mathbb{E}_{X,\tilde{Y}}\left[\mathbb{I}\left[f(X) \neq \tilde{Y}(1 - u^{\text{mle}}) + u^{\text{mle}}(1 - \tilde{Y})\right]\right]$$

$$= \operatorname*{argmin}_{f\in\mathcal{F}}\mathbb{E}_{X,\tilde{Y}}\left[(1 - q_u)\mathbb{I}\left[f(X) \neq \tilde{Y}\right] + q_u\mathbb{I}\left[f(X) = \tilde{Y}\right]\right]$$

We can express the minimizer of the LHS as:

$$f' \in \operatorname*{argmin}_{f\in\mathcal{F}}\mathbb{E}_{X,\tilde{Y}}\left[\mathbb{I}\left[f(X) \neq \tilde{Y}(1 - u^{\text{mle}}) + u^{\text{mle}}(1 - \tilde{Y})\right]\right] \tag{8}$$

$$= \operatorname*{argmin}_{f\in\mathcal{F}}\mathbb{E}_{X,\tilde{Y}}\left[(1 - u^{\text{mle}})\mathbb{I}\left[f(X) \neq \tilde{Y}\right] + u^{\text{mle}}\mathbb{I}\left[f(X) = \tilde{Y}\right]\right] \tag{9}$$

We can denote the minimizer of the RHS:

$$\hat{f} \in \operatorname*{argmin}_{f\in\mathcal{F}}\mathbb{E}_{X,\tilde{Y}}\left[(1 - q_u)\mathbb{I}\left[f(X) \neq \tilde{Y}\right] + q_u\mathbb{I}\left[f(X) = \tilde{Y}\right]\right] \tag{10}$$

Observe that:

$$q_{u|y,\boldsymbol{x}} < 0.5 \implies \hat{f}(X) = \tilde{Y}$$

$$q_{u|y,\boldsymbol{x}} > 0.5 \implies \hat{f}(X) = 1 - Y$$

Thus, we have that $u^{\text{mle}} := \mathbb{I}\left[q_u > 0.5\right] \implies \hat{f} = f'$, as desired. □

## A.2 Results from Section 4

*Proof of Prop. 5.* Denote the noise rate of $\hat{F}(\boldsymbol{x}_i)$ as $e$, that is $\Pr\left(\hat{F}(\boldsymbol{x}_i) \neq y_i\right) = e$.

$$
\begin{aligned}
\mathbb{E}_{Y_i,\hat{F}|\tilde{D}}\left[\mathbb{I}\left[\hat{F}(\boldsymbol{x}_i) \neq \hat{Y}_i\right]\right] &= \mathbb{E}_{Y_i,\hat{F}|\tilde{D}}\left[\mathbb{I}\left[\hat{F}(\boldsymbol{x}_i) \neq \hat{Y}_i \mid \hat{F}(\boldsymbol{x}_i) = y_i\right]\right] \cdot (1-e) \\
&\quad + \mathbb{E}_{Y_i,\hat{F}|\tilde{D}}\left[\mathbb{I}\left[\hat{F}(\boldsymbol{x}_i) \neq \hat{Y}_i \mid \hat{F}(\boldsymbol{x}_i) \neq y_i\right]\right] \cdot e \\
&= \mathbb{E}_{Y_i,\hat{F}|\tilde{D}}\left[\mathbb{I}\left[Y_i \neq y_i\right]\right] \cdot (1-e) + \left(1 - \mathbb{E}_{Y_i,\hat{F}|\tilde{D}}\left[\mathbb{I}\left[\hat{Y}_i \neq y_i\right]\right]\right) \cdot e \\
&= (1-2e) \cdot \mathbb{E}_{Y_i,\hat{F}|\tilde{D}}\left[\mathbb{I}\left[\hat{Y}_i \neq y_i\right]\right] + e
\end{aligned}
$$

When $e < 0.5$, we can claim that the higher the $\mathbb{E}_{Y_i,\hat{F}|\tilde{D}}\left[\mathbb{I}\left[\hat{Y}_i \neq y_i\right]\right]$, the higher the $\mathbb{E}_{Y_i,\hat{F}|\tilde{D}}\left[\mathbb{I}\left[\hat{F}(\boldsymbol{x}_i) \neq \hat{Y}_i\right]\right]$, the ambiguity measure. If we assume that $\mathbb{E}_{Y_i,\hat{F}|\tilde{D}}\left[\mathbb{I}\left[\hat{Y}_i \neq y_i\right]\right]$ is monotonic in the noise rates in $u_i$, which is intuitively true, we then establish that the higher the noise, the higher the ambiguity measure.

$\square$

## A.3 On Choosing an Atypicality Parameter

In Prop. 9, we present an additional bound that can be used to set an atypicality parameter $\epsilon$ to guarantee that the set of plausible draws $\mathcal{F}_\epsilon^{\text{plaus}}$ includes a reference noise draw with high probability.

**Proposition 9.** Given a set of $n_p$ instances $(\boldsymbol{x}, \tilde{y})$ subject to noise rate $p_u$, we determine the minimum $\epsilon$ to ensure that a reference noise draw $\boldsymbol{u}^*$ belongs to the set plausible draws $\mathcal{F}_\epsilon^{\text{plaus}}$ with high probability. That is, with probability at least $1 - \delta$, $\boldsymbol{u}^* \in \mathcal{U}_\epsilon(\tilde{\boldsymbol{y}})$ if $\epsilon$ obeys:

$$
\epsilon \geq \frac{1}{q_{u|\tilde{y}}}\left(\sqrt{\frac{\ln\left(\frac{2}{\delta}\right)}{2n_p}} + |p_u - q_{u|\tilde{y}}|\right).
$$

Here $n_p$ represents the number of instances whose labels are corrupted by the same noise model. For example, under class-level noise, this bound would need to be evaluated separately using the number of instances for each class.

For example, given a dataset with $n = 10,000$ instances under 20% uniform label noise, a practitioner must set $\epsilon \geq 6\%$ to ensure that $\boldsymbol{u}^* \in \mathcal{F}_\epsilon^{\text{plaus}}$ with probability at least 90%.

*Proof of Prop. 9.* Our goal is to show that $\Pr\left(\boldsymbol{u}^* \in \mathcal{U}_\epsilon(\tilde{\boldsymbol{y}})\right) \geq 1 - \delta$. for any given $0 \leq \delta \leq 1$.

The uncertainty set $\mathcal{U}_\epsilon(\tilde{\boldsymbol{y}})$ defined on $p_{u|\tilde{y}}$ is a strongly typical set (see [9]) where the true mean $p_{u|y}$ and the empirical mean is $\hat{p}_u := \frac{1}{n}\sum_{i=1}^n \mathbb{I}\left[u_i = 1\right]$. Thus,

$$
\boldsymbol{u}^* \in \mathcal{U}_\epsilon(\tilde{\boldsymbol{y}}) \iff |\hat{p}_u - p_{u|\tilde{y}}| \leq p_{u|\tilde{y}} \cdot \epsilon \tag{11}
$$

We will derive conditions to satisfy the inequality Eq. (11)

Observe that we can write

$$
\begin{aligned}
|\hat{p}_u - p_{u|\tilde{y}}| &= |(\hat{p}_u - p_u) + (p_u - p_{u|\tilde{y}})| \\
&\leq |\hat{p}_u - p_u| + |p_u - p_{u|\tilde{y}}| \qquad \text{(by the triangle inequality)}
\end{aligned}
$$

We require $|\hat{p}_u - p_{u|\tilde{y}}| \leq p_{u|\tilde{y}} \cdot \epsilon$. Therefore we need $|\hat{p}_u - p_u| + |p_u - p_{u|\tilde{y}}| \leq p_{u|\tilde{y}} \cdot \epsilon$ which implies that $|\hat{p}_u - p_u| \leq p_{u|\tilde{y}} \cdot \epsilon - |p_u - p_{u|\tilde{y}}|$

We observe that $\boldsymbol{u}^*$ is a sequence of bounded, independently sampled random variables. Thus, we can apply Hoeffding's inequality to see that:

$$\Pr\left(|\hat{p}_u - p_u| \geq \alpha\right) \leq 2 \cdot \exp(-2n\alpha^2)$$

Here, $\alpha = p_{u|\tilde{y}} \cdot \epsilon - |p_u - p_{u|\tilde{y}}|$. Rearranging, we have that:

$$\Pr\left(\boldsymbol{u}^* \in \mathcal{U}_\epsilon(\tilde{\boldsymbol{y}})\right) = \Pr\left(|\hat{p}_u - p_u| \leq \alpha\right) \geq 1 - 2 \cdot \exp(-2n\alpha^2)$$
$$= 1 - 2 \cdot \exp(-2n(p_{u|\tilde{y}} \cdot \epsilon - |p_u - p_{u|\tilde{y}}|)^2)$$

We invert the bound to obtain the following statement: with probability at least $1 - \delta$, $\boldsymbol{u}^* \in \mathcal{U}_\epsilon(\tilde{\boldsymbol{y}})$ if the number of samples $n$ obeys:

$$n \geq \frac{-\ln\left(\frac{\delta}{2}\right)}{2(p_{u|\tilde{y}} \cdot \epsilon - |p_u - p_{u|\tilde{y}}|)^2}$$

To conclude the proof, we rearrange for $\epsilon$, that is, given a dataset of $n$ instances, $\boldsymbol{u}^* \in \mathcal{U}_\epsilon(\tilde{\boldsymbol{y}})$ if $\epsilon$ obeys:

$$\epsilon \geq \frac{1}{p_{u|\tilde{y}}} \left( \sqrt{\frac{\ln\left(\frac{2}{\delta}\right)}{2n}} + |p_u - p_{u|\tilde{y}}| \right)$$

$\square$

## B    SUPPORTING MATERIAL FOR SECTION 5 AND SECTION 6

Here we include further details about the datasets used in our experimental results.

### B.1    DATASETS

**lungcancer**    We used a cohort of 120,641 lung cancer patients diagnosed between 2004-2016 who were monitored in the National Cancer Institute SEER study [41] and processed the dataset to match the processing in James et al. [21]. The outcome variable is death within five years from any cause, with 16.9% dying within this period. The cohort includes patients across the USA (California, Georgia, Kentucky, New Jersey, and Louisiana), excluding those lost to follow-up. Features include measures of tumor morphology and histology (e.g., size, metastasis, stage, node count and location), as well as clinical interventions at the time of diagnoses (e.g., surgery, chemotherapy, radiology).

**shock_eicu & shock_mimic**    Cardiogenic shock is an acute cardiac condition where the heart fails to sufficiently pump enough blood [19] leading to under-perfusion of vital organs. These datasets are designed to build algorithms to predict cardiogenic shock in ICU patients as described in Yamga et al. [56]. Both datasets contain identical features, group attributes, and outcome variables but they capture different patient populations. The shock_eicu dataset includes records from the EICU Collaborative Research Database V2.0 [45], while the shock_mimic dataset includes records from the MIMIC-III database [22]. The target variable is whether a patient with cardiogenic shock will die in the ICU. Features include vital signs and routine lab tests (e.g., systolic BP, heart rate, hemoglobin count) collected within 24 hours before the onset of cardiogenic shock.

**mortality**    The Simplified Acute Physiology Score II (SAPS II) score is a risk-score designed to predict the risk of death in ICU patients collected in [28] and used in [51]. The data contains records of 7,797 patients from 137 medical centers in 12 countries. The outcome variable indicates whether a patient dies in the ICU, with 12.8% patient of patients dying. Similar to the other datasets, mortality contains features reflecting comorbidities, vital signs, and lab measurements.

**support**    This dataset comprises 9,105 ICU patients from five U.S. medical centers, collected during 1989-1991 and 1992-1994 [24]. Each record pertains to patients across nine disease categories: acute respiratory failure, chronic obstructive pulmonary disease, congestive heart failure, liver disease, coma, colon cancer, lung cancer, multiple organ system failure with malignancy, and multiple organ system failure with sepsis. The aim is to determine the individual-level 2- and 6-month survival rates based on physiological, demographic, and diagnostic data.

### B.2    ADDITIONAL RESULTS FOR DNN MODELS

In Section 5, we include results for LR models. In this section, we include additional results for DNN models trained on the same unique noise draw as in Section 5.

| Dataset | Metrics | $p_{u\|y=1} = 5\%$ | | $p_{u\|y=1} = 20\%$ | | $p_{u\|y=1} = 40\%$ | |
|---|---|---|---|---|---|---|---|
| | | Ignore | Hedge | Ignore | Hedge | Ignore | Hedge |
| shock_eicu
$n = 3,456$
$d = 104$
Pollard et al. [45] | True Error | 13.3% | 12.8% | 18.6% | 19.2% | 37.5% | 26.2% |
| | Anticipated Error | 14.4% | 14.0% | 20.3% | 22.0% | 25.1% | 26.7% |
| | Regret | 3.0% | 3.0% | 10.1% | 10.1% | 19.7% | 19.7% |
| | Overreliance | 1.1% | 1.0% | 5.3% | 4.7% | 21.4% | 13.1% |
| | Susceptibility | 52.6% | 52.6% | 59.7% | 59.7% | 69.3% | 69.3% |
| shock_mimic
$n = 15,254$
$d = 104$
Johnson et al. [22] | True Error | 15.6% | 15.9% | 18.8% | 16.8% | 32.7% | 22.1% |
| | Anticipated Error | 17.4% | 17.5% | 23.9% | 23.2% | 26.6% | 25.9% |
| | Regret | 2.5% | 2.5% | 10.2% | 10.2% | 19.8% | 19.8% |
| | Overreliance | 0.4% | 0.5% | 3.4% | 2.5% | 17.7% | 10.8% |
| | Susceptibility | 52.5% | 52.5% | 60.2% | 60.2% | 69.8% | 69.8% |
| lungcancer
$n = 62,916$
$d = 40$
NCI [41] | True Error | 29.8% | 29.7% | 31.5% | 30.0% | 37.7% | 29.5% |
| | Anticipated Error | 30.4% | 30.4% | 31.8% | 33.4% | 29.7% | 36.7% |
| | Regret | 2.5% | 2.5% | 10.0% | 10.0% | 19.7% | 19.7% |
| | Overreliance | 1.4% | 1.3% | 7.1% | 5.0% | 19.8% | 9.9% |
| | Susceptibility | 52.7% | 52.7% | 60.2% | 60.2% | 69.9% | 69.9% |
| mortality
$n = 20,334$
$d = 84$
Le Gall et al. [28] | True Error | 17.7% | 17.9% | 19.2% | 18.3% | 24.0% | 18.9% |
| | Anticipated Error | 19.1% | 19.4% | 23.4% | 24.0% | 26.2% | 29.5% |
| | Regret | 2.2% | 2.2% | 9.8% | 9.8% | 19.5% | 19.5% |
| | Overreliance | 0.6% | 0.5% | 3.7% | 2.7% | 11.7% | 6.3% |
| | Susceptibility | 52.2% | 52.2% | 59.8% | 59.8% | 69.5% | 69.5% |
| support
$n = 9,696$
$d = 114$
Knaus et al. [24] | True Error | 28.4% | 28.6% | 31.0% | 30.3% | 39.4% | 35.7% |
| | Anticipated Error | 28.2% | 28.2% | 28.6% | 28.7% | 25.2% | 27.8% |
| | Regret | 2.6% | 2.6% | 10.0% | 10.0% | 19.6% | 19.6% |
| | Overreliance | 2.0% | 2.1% | 8.7% | 8.1% | 22.6% | 19.1% |
| | Susceptibility | 52.6% | 52.6% | 60.0% | 60.0% | 69.6% | 69.6% |

**Table 4:** Overview of performance and regret for DNN model trained on all datasets and training procedures.

