# OpenReview forum: "Regretful Decisions under Label Noise"
_ICLR.cc/2025/Conference — ICLR 2025 Poster_

### Official Review · Reviewer_cRDp · 2024-10-27

**Soundness:** 3
**Presentation:** 3
**Contribution:** 3
**Rating:** 8
**Confidence:** 5

**Summary:**

This work proposes an evaluation framework for noisy label learning methods in terms of "regret," as quantified by the discrepancy between model errors with respect to noisy labels vs. errors with respect to true labels. But regret is not distributed equally in the data: in their own words, "even if we can limit the number of mistakes, we cannot anticipate how they will be assigned over instances that are subject to label noise." The proposed approach takes a generative model of noise to train a set of models on plausible (under the distribution induced by the generative model) clean realizations, and estimates instance-level "regret" accordingly. Empirical results show that a common noisy-label learning baseline and naive approaches (ignore noise) exhibit non-zero regret consistently. A case study on a genomics dataset demonstrates the practical utility of the approach by leveraging an instance-level ambiguity measure derived from regret to abstain from low-confidence predictions.

**Strengths:**

* This is a very well-written paper. The prose is clear and the technical aspects of the problem motivation are well-defined and explained concisely.
* The proposed approach is very simple and the theoretical results are intuitive, but backed by rigorous theoretical and empirical analyses.
* Rather than assuming completely random models of noise, the proposed approach is adaptable to arbitrary generative models of noise.

**Weaknesses:**

* [W1 — knowing the true noise model]: The proposed approach requires knowledge of a full generative noise model of $(U, X, Y)$. It is not clear where this would come from in practice. This weakness is somewhat mitigated by the discussion at the end of Section 3 and empirical results showing robustness of the proposed approach to noise model misspecification, but building more formal machinery to characterize the sensitivity of the approach to noise model misspecification would strengthen the paper.
* [W2] Proposition 4 provides the motivation for the proposed approach — using the generative model of noise, sample plausible realizations of the clean dataset. But the variance of the posterior could be extremely high — even with the $\varepsilon$-plausibility constraint (Def. 7), this could still yield high-variance regret/ambiguity averages.
* [W3] I'm unsure about the usefulness of Prop. 5, which "implies that we can only expect hedging to learn a model that does not assign
unanticipated mistakes when $\mathbf{u}\_{mle} = \mathbf{u}\_{true}$. I read this as "models will overfit in finite samples to the observed noise draw rather than the true noise draw," which is intuitive. But if regret grows very, very slowly in $|\mathbf{u}\_{mle} - \mathbf{u}\_{true}|$ (any measure of distance between the two, to abuse some notation) — then it seems like this effect is not an issue.
* [W4 — minor] The presentation of empirical results could be improved. Table 3 is very large, and it's hard for me to parse what I'm looking for. Similarly, Figures 3 and 4 could be designed a little more informatively — specifically, the caption should include a statement about why the proposed approach is "better" (e.g., our approach has X property, while the standard approach ... ).

**Questions:**

* Re: [W1] — I would love to hear any thoughts on the robustness of the proposed approach to noise model misspecification from a theoretical perspective.
* Re: [W2] — I would love to hear any commentary on how high-variance in the noise posterior could negatively affect the proposed approach.
* Re: [W3] — Are small violations of the $\mathbf{u}\_{mle} = \mathbf{u}\_{true}$ condition (Prop. 5) truly "problematic?" Is there an example to demonstrate this?

**Other questions/suggestions**
* Did the authors consider looking at metrics beyond expected regret/ambiguity (e.g., worst-case over $\varepsilon$-plausible models)?
* The noisy label evaluated in the experiments is >10 years old; while the value of the approach isn't based on which underlying noisy label learning method is under evaluation, it might be more salient to the noisy-label learning community to test a more recent suite of methods + different noise models. For example, [some](https://arxiv.org/abs/1809.03207) [methods](https://arxiv.org/abs/2406.18865) specify a full generative model and cast the clean label as a latent variable, while [other](https://arxiv.org/abs/2002.07394) [approaches](https://arxiv.org/abs/1910.01842) filter out examples flagged as noisy (according to some rule) in the learning process. Given the plethora of assumptions/noise models in the literature, I wouldn't be shocked if there is systematic variation in errors across methods.

**Minor Suggestions**
* In Table 2, $\hat{\mu}(x)$ is defined as the median, but in Eq. (8), it is defined as the mean — I suggest making the definition consistent.
* The proof of Prop. 4 in Appendix A is a little unclear: there are also some typographical inconsistencies (math mode vs. regular font), and I think $f(X)$ is mistakenly written as $X$ in one of the loss terms as L725-726. I was unable to replicate the final step, but this is likely since I had a hard time following the parentheses/whether each line was a continuation of the previous. Could this be clarified? I believe the result, since it seems to be essentially a result of the form E_{noise}[estimand] = estimand as is common in the noisy-label learning literature.
* Is Prop. 9 (Appendix A only) not simply a restatement of Lemma 1 from [Learning with Noisy Labels, Natarajan et al., NeurIPS '13](https://proceedings.neurips.cc/paper_files/paper/2013/file/3871bd64012152bfb53fdf04b401193f-Paper.pdf)? If so, the proof can be omitted and replaced with the relevant citation.
* Prop 10. and 11 appear to be standard applications of a weak law of large numbers + Hoeffding. If they're not referenced in the main text, consider removal.

---

> ### Author Response · Authors · 2024-11-22
> **Response to Reviewer cRDp (I)**
>
> Thanks for your time and feedback! We very much appreciate the detailed read and suggestions. We've addressed most of the questions and comments below. If there is anything else, please let us know!
>
> > **Did the authors consider looking at metrics beyond expected regret/ambiguity (e.g., worst-case over ε-plausible models)?**
> Yes! We can compare the performance of our Ambiguity measure against some basic alternatives.
>
> In this case, we ran a simple example where we fit a logistic regression model on the shock_eicu dataset under class-conditional noise. We then evaluated how well we can abstain from regretful decisions in a standard selective classification setup. In this case, we evaluate the "confidence" of each instance using three carefully selected measures.
>
> * $\textrm{Ambiguity}(x_i) = \sum_k 1[f^k(x_i) ≠ \hat{y}_i^k]$. This is the current measure of ambiguity. It measures the fraction of plausible models $f^k$ that make a mistake on $x_i$ with respect to the plausible labels $ \hat{y}_i^k$.
>
> * $\textrm{Alternative}(f, x_i) = \sum_k 1[f(x_i) ≠  \hat{y}_i^k]$. This is an alternative measure that we could use if we did not wish to train additional models. It measures the accuracy of the model for a given model "f" with respect to plausible labels - comparing against this measure can highlight the impact of training models.
>
> * $\textrm{Confidence}(x_i) = p(\tilde{y}_i \mid x_i)$. This is the standard softmax score. It represents a baseline measure that we would use for selective classification in regimes where there is no label noise (or we ignore it).
>
> We provide a figure that highlights how these measures perform in our [anonymous repository](https://anonymous.4open.science/r/noise_multiplicity_iclr2025/abstain_rebuttal.pdf). Here, we abstain on regretful instances using the threshold rule: $\mathbb{I} [\textnormal{measure} ({x_i}) \leq \tau]$, where ‘measure’ refers to any of the measures listed above, and vary the threshold $\tau$. We then plot the selective regret when we only assign predictions to instances with confidence $\geq$ threshold.
>
> Our results in this experiment (and others) provide some motivation for our measure. We see:
>
> Our metric outperforms standard measures of confidence.
> Measures that are based on confidence account for the "dataset" (in this case, we know that some points are not subject to regret at all)
> We can achieve slightly better performance by training models. ML algorithms have *some* degree of robustness to label noise. For example, training a DNN with 20% noisy labels can often lead to a model with $\geq 85$% accuracy (see e.g.,  Table 1 [in this paper from ICLR 2021](https://arxiv.org/pdf/2010.02347)). Our approach uses the inherent robustness of model training and provides a cleaner “proxy” prediction and therefore less variance in computing a confidence measure.
>
> Looking forward, we plan to integrate the response above into Section 3. We will also discuss the mechanisms that lead to gains in our experimental section.
>
> > **Building more formal machinery to characterize the sensitivity of the approach to noise model misspecification would strengthen the paper**
>
> We agree that this is one of the weaknesses of the current approach. In principle, our machinery could be generalized to account for misspecification. Specifically, we can train a set of plausible classification models for a *family* of noise models. In this simplest case, we could train this set using a hierarchical approach where we first sample the noise model and then run the existing procedure.
>
> We considered including a simple proof-of-concept demonstration for this approach in the current paper, but decided against it for editorial reasons. First, we realized that we may be able to avoid the "hierarchical approach" for some salient classes of noise models (e.g., by calling our procedure with a larger value of $\epsilon$ that accounts for atypicality and misspecification). Second, we thought that it would be important to pair any technique with some empirical evidence that it works reliably (which requires space and detracts from the main text).
>
> We've tried to be as explicit as possible about this assumption and to discuss its potential limitations. Looking forward, we can include references to methods that estimate a noise model and include a brief version of the approach described above if you think it would add value. Let us know!
>
> >  **I would love to hear any thoughts on the robustness of the proposed approach to noise model misspecification from a theoretical perspective.**
>
>
> > **I would love to hear any commentary on how high-variance in the noise posterior could negatively affect the proposed approach.**
>
> As the noise posterior is a Bernoulli random variable, it is possible to show that for any such distribution, the variance is within [0, 0.25]. Combined with the choice of $\epsilon$ in the set of plausible draws, variance is controlled in a principled way.

---

> ### Author Response · Authors · 2024-11-22
> **Response to Reviewer cRDp (II)**
>
> > **Are small violations of the umle = utrue condition... truly "problematic?"**
>
> Thanks for bringing this up! To give some context, the two points we wanted to make by presenting this results are:
>
> Hedging behaves in a way that is intuitive and interpretable: hedging optimizes for a maximum-likelihood noise draw
> Show that regret is inevitable as the true noise draw is unlikely to be equal to the maximum-likelihood noise draw. Where there are disagreements, regret will arise.
>
> In practice, you're right that small violations in $u_{mle} \neq u_{true}$ may not be problematic. Our point here is that *whenever*  $u_{mle} \neq u_{true}$ then we are bound to experience regret (i.e., "hedging" can help with respect to error but "regret" still remains.). Even if violations are small, these are still instances (or individuals) that are subject to a lottery of mistakes - which may have consequences depending on how the predictions are used. Note that this result is now renamed to Prop 4 in the revision.
>
> >  **it might be more salient to the noisy-label learning community to test a more recent suite of methods + different noise models.**
>
> We agree. We plan to include results for one more method in the Appendix as part of our response to another reviewer (the “forward” loss-correction as defined in Theorem 2 in [Patrini et al. 2017](https://arxiv.org/abs/1609.03683)). In general, we're happy to add more methods so long as they cover different strategies to correct for label noise. As of now, our plan is to implement a recent method that works without the need for a noise model: [Peer Loss](https://arxiv.org/abs/1910.03231), unless you have other suggestions!
>
> > **Median Ambiguity**
>
> We think this was a misunderstanding. The mean value is the ambiguity estimate for a single instance $x_i$ : $\hat{\mu}({x\_i}) = \sum\_{k=1}^m \mathbb{I}[f^k(x\_i) \neq y\_i^k]\$
> The median value in Table 2 is the median estimate ambiguity for all instances in the dataset: $\textrm{Median}_{i=1 \dots n} \hat{\mu} (x_i)$ – we've now clarified this in the text!
>
> > **Prop 4**
>
> This should have been clearer and cleaner. We've reorganized the proof so that it should be easier to follow. Note that this is now listed as "Prop 3" in the revision.
>
> > **Prop 9**
>
> The result is indeed equivalent to Lemma 1 in Natarajan et al. We've referenced their work so that credit goes where it's due. We've left the result in the Appendix for the sake of completeness.
>
> > **Prop 10–11**
>
> Removed!
>
> > **The presentation of empirical results could be improved**
>
> Thank you for flagging this! We have added details to the Figure captions to highlight key takeaways for the reader. We will work to find a way to better present the content of Table 3 with reduced volume.

---

> > ### Comment · Reviewer_cRDp · 2024-11-22
> >
> > Thanks for the detailed response. I think I am okay with the limitations on the theoretical side. Nice insights about how to extend the approach to account for noise model misspecification as well; i.e., it seems like it's just another layer of "uncertainty" that can be added to the approach.
> >
> > Re: additional approaches, I agree that peer loss is a great choice (and has the bonus of being fairly easy to implement). The categories of approaches I suggested (i.e., methods to "filter" out examples flagged as noisy + methods that assume some generative model of noise) are a little more involved but could be of interest as they indeed represent different strategies to address noisy labels. In particular, the "filtering" approaches might be an interesting comparison, since it seems to map on to the paper's notion of "regretful" predictions.
> >
> > Ultimately these are minor comments — I keep my score and continue to advocate for acceptance of this paper.

---

### Official Review · Reviewer_mXWQ · 2024-11-01

**Soundness:** 3
**Presentation:** 2
**Contribution:** 3
**Rating:** 8
**Confidence:** 4

**Summary:**

The authors tackle the situation where observations come with label noise. They introduce a criterion (regret) which measures when the prediction errors disagree when the model is computed with  noisy observation \tile{y} and not noisy y. They develop a new method that estimate the posterior distribution of the possible noise and try to sample these observations. Hence if the distribution of the noise is well chosen, it becomes possible to construct set of plausible models and thus detect zones for which the uncertainty is above a certain level. The paper develop a new theory and provides sound mathematical proofs and simulations.

**Strengths:**

The point of view which is developed is interesting and is a valuable contribution.
The ideas are straightforward once the frame is set : defining the regret and then plausible sets minimizing the regret on epsilon-plausible datasets.
Experiments are convincing.
A whole section is devoted to the theoretical analysis of the results. Proposition 12 proposes the statistical guarantees of the methos.

**Weaknesses:**

I found the paper sometimes difficult to read and some sentences are difficult to understand
l159 : "a practitioner may be able they expect ... " I can not understand what the authors mean.
l162 : the definition of the regret is fuzzy with some words that are not properly defined . "anticipated" mistake. What does it mean since you compare the error with labels with noise and labels without ?
l172 the paper should be self-contained if possible, so epxlain the comparison with \tilde{l}_{0,1}
l182 : what is " :-= " ?
l215: sometimes you use words that have a mathematical meaning : "most likely to flip" for instance

**Questions:**

can you define \tile{l}_{0,1}

---

> ### Author Response · Authors · 2024-11-22
> **Response to Reviewer mXWQ**
>
> Thank you for your time and feedback! We are pleased to see that you found our point of view interesting and a valuable contribution to the field with convincing experiments. We also appreciate the feedback you have pointed out, and hope to clarify these questions below:
>
> > **I found the paper sometimes difficult to read and some sentences are difficult to understand**
>
> Thank you for pointing this out! We've uploaded a new version that addresses them all and contains several other improvements to the writing. If there is anything else that was confusing, please let us know and we will seek to address it.
>
> > **Can you define \tile{l}_{0,1}?**
> We agree! To restate here:
> $\tilde{\ell}_{0,1}: X, \tilde{Y} \to Y$ is an instance-based loss function that is a popular approach to dealing with noisy labels first described in [this NeurIPS paper](https://www.ambujtewari.com/research/natarajan13learning.pdf)]. Consider a task where we have a class-conditional noise model where the noise is generated according to $Pr(U=1 \mid Y)$, for example.
>
> If we define
> $\tilde{\ell}_{0,1} (f(x), \tilde{y}) := \frac{(1-Pr(U=1 \mid Y=1-y))\ell(f(x), \tilde{y}) - Pr(U=1 \mid Y=y)\ell(f(x), 1-\tilde{y}) }{1-Pr(U=1 \mid Y=0) - Pr(U=1 \mid Y=1)} $
>
> where $\ell(\cdot)$ is any loss function (e.g., cross entropy), then the loss function $\tilde{\ell}_{0,1}$ is unbiased in the sense that:
> \$\mathbb{E}\_U \[ \tilde{\ell}\_{0,1}(X, \tilde{Y}) \] = \ell\_{0,1}\$
>
> That is to say, we can use $\tilde{\ell}_{0,1}$ to learn from a noisy-data distribution and, under expectation, this coincides with the same loss as if we had access to clean labels! This loss function can be used to learn a classifier robust to label noise.
>
> We chose this since it represents the simplest version of hedging that we can think of that is widely used to handle label noise. We've now updated this in the text.
>
> > **[What is] an "anticipated" mistake. What does it mean since you compare the error with labels with noise and labels without ?**
>
> Yes, we are happy to explain. The idea of an “anticipated” mistake, $e^{pred}(\cdot)$ , is a practitioner’s intuition about whether a given prediction is correct or not:
> If the practitioner is ignoring noise then: $e^{pred}(f(x), \tilde{y}) =\mathbb{I} [f(x) \neq \tilde{y}] $, an unmodified zero-one loss with noisy labels
> If the practitioner is accounting for noise then:  $e^{pred}(f(x), \tilde{y}) =\tilde{\ell}_{0,1} (f(x), \tilde{y}) $, which can be any loss function suitable for learning with noisy labels, such as the one described in the response above (see e.g., [this NeurIPS paper](https://www.ambujtewari.com/research/natarajan13learning.pdf)])
> We use the idea of anticipated mistake as it can encapsulate any type of loss function that a practitioner may use to evaluate their model’s performance. Using this idea, we are able to define how regret arises because of mistakes in anticipation - a practitioner not knowing where their model is making mistakes.
>
> > **l159 : "a practitioner may be able they expect ... "**
> > **l162 : the definition of the regret is fuzzy with some words that are not properly defined .**
> > **l215: sometimes you use words that have a mathematical meaning : "most likely to flip" for instance**
>
> Most of these were unfortunate typos. We've fixed all of these in our revision. If there is anything else that was confusing, please let us know and we will seek to address it.

---

### Official Review · Reviewer_gw9a · 2024-11-02

**Soundness:** 2
**Presentation:** 2
**Contribution:** 2
**Rating:** 6
**Confidence:** 3

**Summary:**

This paper examines the problem of learning from noisy labels by addressing instance-level noise. The main contribution is the insight that a method performing well over the population can still lead to errors at the instance level. The paper introduces the concept of "regret" to characterize this phenomenon and proposes a method to mitigate the regret caused by randomness sampling multiple plausible noisy label draws. Theoretical analysis and experiments are presented to validate the proposed approach.

**Strengths:**

+ This paper addresses an important yet challenging task: learning from instance-level label noise.
+ Theoretical analysis and experiments are conducted to validate the proposed method.

**Weaknesses:**

- **Unclear Benchmark Algorithm**: One of my main concerns is the paper's clarity, particularly regarding the introduction of the benchmark algorithm critiqued in Section 2. It appears the paper intends to use a noise-tolerant method, such as that of Natarajan et al. [37], as a benchmark. However, by Proposition 5, the algorithm under discussion seems to actually refer to a different approach (let's call it Benchmark 2). In Benchmark 2, an implicit noise draw $\mathbf{u}^{\mathrm{mle}}$ is generated, $y_i$ is recovered using this noise draw, and then ERM is performed to train the model. To me, this algorithm (Benchmark 2) differs from the method in [37], especially in terms of instance-level performance. Therefore, it is less convincing that the criticisms for Benchmark 2 are applicable to the noise-tolerant method in [37]

- **Clarity on Notation**: I find the notation in Section 2 somewhat confusing, particularly in distinguishing which variables are random and which are deterministic. Based on the discussion in lines 130-135, it appears that $ y_i $ is deterministic, while $ U_i $ and $ \tilde{y}_i $ are random variables generated based on $ y_i $. However, I struggle to interpret the equation in line 173, as it seems $ y_i(U_i) $ is simply a deterministic value, making it challenging to see how it could be compared in inequality to a random variable.

- **Regarding Proposition 4 and its Proof**: It would be helpful if the authors provided a clearer explanation of which random variables the expectation is taken over. In the proof, it appears that the expectation is taken over $ X, \tilde{Y} $, and $ U $ while this is not mentioned in the main text. Additionally, I find it difficult to follow the reasoning in lines 734-744; the conclusion seems to rely on $ E_{X, \tilde{Y}, U}[\text{Regret}] $, yet the analysis is conducted for $ E_{X, Y, U}[\text{Regret}] $. Finally, the last lines indicate only that $ E_{X, \tilde{Y}, U}[\text{Regret}] > 0 $, but it is unclear how this leads to the conclusion stated in Proposition 4.

- **Strong Assumption**: The proposed method in Section 3 requires knowledge of $ P(U = 1 \vert X, Y) $. This is somewhat a strong assumption to me, as accurately estimating this value is generally challenging.

- **Insufficient Experiments**: It appears that the paper does not compare the proposed method with others in the literature. At a minimum, it would be beneficial to include a comparison with the noise-tolerant method [37] under the condition that $ P(U = 1 \vert X, Y) $ is known. Although [37] is designed for class-dependent noise, with knowledge of  $P(U = 1 \vert X, Y)$, extending it to handle instance-level noise should not be too challenging.

**Questions:**

- Could you elaborate further on the benchmark algorithm discussed in the paper and clarify its relationship with [37]?
- Could you provide an additional explanation regarding the proof of Proposition 4, particularly addressing the concerns mentioned in the weaknesses above?
- Could you include a performance comparison with other methods in the literature?

---

> ### Author Response · Authors · 2024-11-22
> **Response to Reviewer gw9a (I)**
>
> Thank you for your time and feedback! We are pleased to see that our paper addresses an important task, and we appreciate the feedback you have pointed out. That being said, we believer there might be a few misunderstandings we should clarify -  we hope to clarify these concerns below:
>
> > **It would be beneficial to include a comparison with the noise-tolerant method [Natarajan et al. 2013] under the condition that  P(U=1|X,Y) is known.**
>
> > **The algorithm under discussion seems to actually refer to a different approach (let's call it Benchmark 2). In Benchmark 2, an implicit noise draw umle is generated, yi  is recovered using this noise draw, and then ERM is performed to train the model.**
>
> > **Could you elaborate further on the benchmark algorithm... [and its relationship] with Natarajan et al. 2013?**
>
> We think that there is a misunderstanding! To be clear:
>
> The benchmark algorithm that we are using (i.e., ‘Hedge’ in Table 2) is exactly the same algorithm that is described in Natarajan et al. That is, we solving the ERM problem: $f \in \arg \min \sum_{i=1}^n [\tilde{\ell} (f(x), \tilde{y})]$ where $\tilde{\ell}_{0,1} (f(x), \tilde{y}) := \frac{(1-Pr(U=1 \mid Y=1-y))\ell(f(x), \tilde{y}) - Pr(U=1 \mid Y=y)\ell(f(x), 1-\tilde{y}) }{1-Pr(U=1 \mid Y=0) - Pr(U=1 \mid Y=1)} $, is the noise-tolerant loss function defined in Lemma 1 of [ Natarajan et al 2013](https://www.ambujtewari.com/research/natarajan13learning.pdf)]
>
> We agree that a benchmark algorithm (i.e., "Benchmark 2") designed using the result in Prop 5 would not reflect a meaningful comparison. In addition to the reasons that you describe, part of the reason why this would not work is because it is not well specified: (1) Natarajan et al. uses convex surrogate loss functions; (2) $u_{mle}$ might not be unique;
>
> Prop 5 is a simple theoretical result. We include it because it provides some simple intuition for how a hedging algorithm behaves.
>
> > **The proposed method in Section 3 requires knowledge of P(U=1|X,Y)**
>
> Thanks for bringing this up!
>
> This assumption could have been clearer and we now see how it may have been confusing. To be clear, the proposed method will require knowledge of different quantities. This quantity depends on the noise model:
>
> If we have *uniform noise*, we require knowledge of $\textrm{Pr}(U = 1)$
>
> If we have *class dependent noise*, we require knowledge of $\textrm{Pr}(U = 1 | Y)$
>
> If we have *group dependent noise*, we require knowledge of $\textrm{Pr}(U = 1 | Y, G)$
>
> If we have *instance-dependent noise*, we require knowledge of $\textrm{Pr}(U = 1 | Y, X)$
>
> In our original submission, we wrote $\textrm{Pr}(U=1|X, Y)$ because we wanted to show that our method could naturally handle the "most complex" noise model. Looking back, this may have inadvertently made it seem like we would always require this assumption. We have updated the text to make this clear.
>
> > **This is somewhat a strong assumption to me, as accurately estimating $\textrm{Pr}(U = 1 | Y, X)$ is generally challenging.**
>
> We agree that this is difficult in general. We do note that it is sometimes possible to estimate $\textrm{Pr}(U = 1 | Y, X)$. In the demonstration, for example, we consider a discovery task where we are predicting the outcome of an in-vitro experiment. In this case, we have an instance-dependent noise where we can estimate the $\textrm{Pr}(U = 1 | Y, X)$ as follows:
>  $\textrm{Pr}(U = 1 | Y = 0, X) $ denotes the Type 1 error i.e., rejecting the null hypothesis when it is true. The standard Type 1 error rate scientists use in reporting their findings is 5% (e.g., when an experiment is claimed to be statistically significant $p < 0.05$)
>
>  $\textrm{Pr}(U = 1 | Y = 1, X) = $ denotes the Type 2 error i.e, failing to reject a null hypothesis that is actually false. Type 2 error is inversely related to the statistical power of an experiment, e.g., it is reduced with a large sample size.
> In this case, the estimation is possible because each "instance" represents the outcome of an experiment where we have multiple trials and corresponding Type 1 and Type 2 error.

---

> > ### Author Response · Authors · 2024-11-22
> > **Response to Reviewer gw9a (II)**
> >
> > > **Could you include a performance comparison with other methods in the literature?**
> >
> > Sure, we would be happy to include another method!
> >
> > We will plan to include another noise-tolerant method by [Patrini et al. 2017](https://arxiv.org/abs/1609.03683) in the Appendix. The results are by and large similar to those for Natarajan et al.–  i.e., the method performs well at a population-level but is unable to reduce regret at the instance-level. This method as well as the one we already included are regarded as gold-standard methods in the field in that they possess statistical guarantees of performance. If you'd like for us to include comparisons to another method, please let us know. We'll try to run them before the end of the discussion period or a potential camera ready.
> >
> > One point we'd like to make: we don't think that including additional methods to the experiments will add value since our goal is to not to compare "performance" but to discuss their effects at the instance level. In this case, most other methods in the literature will assign unpredictable mistakes at the instance level. With that being said, we're happy to add performance comparisons with other methods that could mitigate these effects.
> >
> >
> > > **I struggle to interpret the equation in line 173, as it seems yi(Ui) is simply a deterministic value, making it challenging to see how it could be compared in inequality to a random variable.**
> >
> > This may be a bit confusing since we were assuming the perspective of a practitioner. In this case our intent was to describe the following scenario:
> > * The practitioner knows $\tilde{y}_i$
> >
> > * The practitioner does not know the true label $y_i$
> >
> > This can be written as either $Y_i$ (random variable) or $y(U_i) = y_i \oplus U_i$.
> >
> > Because $y_i$ is the function of an observed value $\tilde{y}_i$ and a random variable $U_i$, it is itself a random variable. However, we would be happy to use a different notation if you think it would be clearer. Please let us know!
> >
> > > **Could you provide an additional explanation regarding the proof of Proposition 4, particularly addressing the concerns mentioned in the weaknesses above?**
> >
> > Sure! This should have been clearer and cleaner. We've reorganized the proof so that it should be easier to follow. Note that this is now listed as "Prop 3" in the revision. This result shows that Regret coincides with the noise rate, even if a noise-tolerant loss function (e.g., Natarajan et al.) can achieve zero error.
> >
> > Please let us know if this makes sense!

---

> > > ### Comment · Reviewer_gw9a · 2024-11-26
> > >
> > > I appreciate the authors' feedback. Upon another check, I agree that the noise-correction method is also subject to instance-level mistakes. Since the main concern has been addressed and the authors have reformulated the notations and improved the proof, I am increasing my score to 6.

---

### Official Review · Reviewer_tJN2 · 2024-11-03

**Soundness:** 2
**Presentation:** 3
**Contribution:** 2
**Rating:** 5
**Confidence:** 5

**Summary:**

The authors introduce the notion of regret when learning from a dataset that is subject to label noise. The authors point out that standard learning approaches typically target a notion of “average” loss or “average” risk over the population and cannot provide instance level guarantees. One way to identify that the model may make a mistake is to have access to clean labels, but this is often infeasible in practice. As a result, the authors propose to simulate “clean” datasets by assuming a noise model, simulating noise from that noise model, and then backing out a clean dataset from the noisy dataset and the sampled noise. Then the authors define a notion of ambiguity based on models trained on various plausible clean datasets.

**Strengths:**

This paper emphasizes that standard machine learning methods target some notion of “average loss” or “average risk” but that does not provide guarantees on performance for an individual instance, which is an important point. In particular, I enjoyed lines 186-188 “we cannot anticipate how [mistakes] will be assigned over instances that are subject to label noise. In this case, each instance where [there is a nonzero probability of a label flip] is subjected to a lottery of mistakes.”

The idea of constructing multiple plausible clean datasets from a noisy one is interesting, and seems very reminiscent of distributionally robust optimization (the idea constructing the set of plausible noise draws seems related to constructing a robustness set over distributions). It might be worthwhile to consider what connections there are between constructing the set of plausible noise draws and a robustness set.

**Weaknesses:**

- It would be helpful if the authors could provide additional explanation on how the notion of regret differs from standard classification accuracy, and why it is useful.

- A key limitation of the approach is that it requires the machine learning practitioner to specify a reasonable noise model.

- The justification for restricting the sampled noise draws to a set of “plausible” noise draws is not clear to me. Why can’t we just account for the fact that each noise draw has a different likelihood?

- The ambiguity quantity is not well-motivated. Why is it defined as the fraction of misclassifications across the cleaned datasets? How do the authors intend this ambiguity quantity to be used? Under what conditions is ambiguity equal to zero?

**Questions:**

Why do we write $y_{i}(U_{i})$ in the definition of $U_{i}$, from what I recall, $U_{i}$ is generated given $y_{i}$, so it is a bit confusing to think of $y_{i}$ as a function of $U_{i}$.

---

> ### Author Response · Authors · 2024-11-22
> **Response to Reviewer tJN2 (I)**
>
> Thank you for your time and feedback! We are pleased to see that you found our problem setting important and our methods interesting, we appreciate the feedback you have pointed out. We hope to clarify these concerns below:
>
> > **It would be helpful if the authors could provide additional explanation on how the notion of regret differs from standard classification accuracy, and why it is useful.**
>
> Sure! Standard classification accuracy would capture how many mistakes we are making on a clean dataset. At the individual level, given a prediction on an instance and a label, we know if we are making a mistake or not, i.e. $\textnormal{mistake} = f(x_i) \neq y_i$.
>
> Regret captures how many *unanticipated* mistakes we make when learning from label noise due to the inherent uncertainty in the labels. In this case, we may *think* a model is correct but is in reality a mistake. Alternatively, there are cases where we may *think* that we are making a mistake but the model is in fact correct. For example, if we are ignoring label noise, we may over rely on *incorrect* predictions because of our inability to anticipate mistakes.
>
> The notion of regret is useful because it can help us identify instances where our ability to anticipate mistakes is fundamentally broken. In scenarios where predictions may impact critical decisions (e.g., healthcare), this can be particularly dangerous. In Section 4 and 5, we provide real-world demonstrations on how we can reduce regret via selective classification.
>
> > **How do the authors intend this ambiguity quantity to be used?**
>
> We expect that ambiguity can be used as a confidence score that captures the likelihood of making a mistake. In practice, we would expect that this quantity can be used as a plug-in estimate for approaches such as selective classification (where we can abstain from regretful decisions), or active learning (where we could clean the labels of regretful instances).
>
> In general, these strategies provide a way to use models without incurring regret given a suitable confidence measure. In this case, we focus on "selective classification" as a running example since active learning may not always be possible in this regime. As we show in our experiments and demonstration, Ambiguity works quite well as a plug-in confidence estimate in this regime – i.e., when we abstain from predictions using Ambiguity, we find that we can effectively improve selective error and reduce the rate of unanticipated mistakes.

---

> ### Author Response · Authors · 2024-11-22
> **Response to Reviewer tJN2 (II)**
>
> > **The ambiguity quantity is not well-motivated...**
>
> We are assuming that this refers to motivation for "why it works" as a confidence measure rather than "evidence that it works." We have some intuition that we can point to to support this. In short, the *effective* noise for an instance will depend on three factors:
>
> (i) the noise model;
>
> (ii) the distribution of noisy labels in the training data;
>
> (iii) the ability to resolve label noise through training.
>
> Given a noise model, we want to identify a confidence measure that satisfies (ii) and (iii). The motivation for (ii) is straightforward - specifically, different points can have different susceptibility to label noise. For example, in the setting of class-dependent noise where only one class experiences noise, we know that some points will never experience regret. Our reasoning for (iii) stems from model training having *some* degree of tolerance to noise. For example, training a DNN with 20% noisy labels can often lead to a model with $\geq 85$% accuracy (see e.g., Table 1 in this paper from ICLR 2021). This indicates that retraining confers some degree of stability in model predictions on individual instances. Our approach leverages the inherent noise robustness of model training and provides a cleaner “proxy” prediction and therefore less variance in computing a confidence measure.
>
> Stepping back, however, maybe the best way that we can motivate this quantity is through a simple example where we compare how it works against some basic alternatives.
>
> In this case, we ran a simple example where we fit a logistic regression model on the shock_eicu dataset under class-conditional noise. We then evaluated how well we can abstain from regretful decisions in a standard selective classification setup. In this case, we evaluate the "confidence" of each instance using three carefully selected measures.
>
> * $\textrm{Ambiguity}(x_i) = \sum_k 1[f^k(x_i) ≠ \hat{y}_i^k]$. This is the current measure of ambiguity. It measures the fraction of plausible models $f^k$ that make a mistake on $x_i$ with respect to the plausible labels $ \hat{y}_i^k$.
>
> * $\textrm{Alternative}(f, x_i) = \sum_k 1[f(x_i) ≠  \hat{y}_i^k]$. This is an alternative measure that we could use if we did not wish to train additional models. It measures the accuracy of the model for a given model "f" with respect to plausible labels - comparing against this measure can highlight the impact of training models.
>
> * $\textrm{Confidence}(x_i) = p(\tilde{y}_i \mid x_i)$. This is the standard softmax score. It represents a baseline measure that we would use for selective classification in regimes where there is no label noise (or we ignore it).
>
> We provide a figure that highlights how these measures perform in our [anonymous repository](https://anonymous.4open.science/r/noise_multiplicity_iclr2025/abstain_rebuttal.pdf). Here, we abstain on regretful instances using the threshold rule: $\mathbb{I} [\textnormal{mistake}({x_i})\leq \tau]$, where ‘measure’ refers to any of the measures listed above, and vary the threshold $\tau$. We then plot the selective regret when we only assign predictions to instances with confidence $\geq$ threshold.
>
> Our results in this experiment (and others) further justify our measure. We see our metric outperforms standard measures of confidence. Measures that are based on confidence account for the "dataset" (in this case, we know that some points are not subject to regret at all). Our approach achieves slightly better performance by training models, by leveraging the inherent noise robustness of model training.
>
> Looking forward, we plan to integrate the response above into Section 3. We will also discuss the mechanisms that lead to gains in our experimental section.
>
> >  **Under what conditions is ambiguity equal to zero?**
> Ambiguity = 0 when there is no noise. This is a property that should be clear for all, given our theoretical results showing the relationship between noise and regret. We can see from Eq 6 that this condition is met, because under no noise, the ‘cleaned’ label $\hat{y}$ would never differ from the noisy label $\tilde{y}$.
>
> > **I enjoyed lines 186-188 “we cannot anticipate how [mistakes] will be assigned over instances that are subject to label noise. In this case, each instance where [there is a nonzero probability of a label flip] is subjected to a lottery of mistakes.”**
>
> Thank you! We can refer to this as a new metric ("Susceptibility") and will report in our experiments if you think it would be useful. This metric can be used to quantify how many individuals in a dataset are subject to a lottery of mistakes.

---

> ### Author Response · Authors · 2024-11-22
> **Response to Reviewer tJN2 (III)**
>
> > **The justification for restricting the sampled noise draws to a set of “plausible” noise draws is not clear to me. Why can’t we just account for the fact that each noise draw has a different likelihood?**
>
> You’re right - this would be possible! The motivation for our use of “plausibility” stems from tasks where we would like to estimate ambiguity using a small number of draws (e.g., m= 300). If we are working with a DNN, then we might have to re-train m=300 DNNs. In a case like this, there is some potential to encounter atypical draws - outlier draws that are not representative of the noise model (e.g., under a 20% noise model, only drawing 5% noise). We could avoid these by allowing  $m \rightarrow \inf$ - however, this is not always feasible (due to e.g., finite compute, prohibitive time-constraints). Alternatively, we can use plausible draws, which gives us a principled way to restrict the noise draws to those that are most representative of the noise-posterior.
>
> > **A key limitation of the approach is that it requires the machine learning practitioner to specify a reasonable noise model.**
>
> Yes this is correct. We agree that this is an inherent assumption and a potential limitation, however it is not a fatal flaw. As we discuss in our response to reviewer cRDP, this is something that we can address using our framework but would require a standalone paper (see our response to reviewer cRDP).
>
> We've tried to be as explicit as possible about this assumption and to discuss its potential limitations. Looking forward, we can include references to existing methods that estimate a noise model.
>
> This is a part of the reason why we included Figure 2 – where we consider a noisy dataset with a true noise rate of 20%. We then apply our procedure to estimate Ambiguity under misspecified noise models with noise rates between 1% to 40% (i.e., what happens if a practitioner under- or overestimates the true noise). In this case, we observe that severe misspecification can affect our confidence estimates. In practice, however, we find that this effect is moderated as our procedure is stable at reducing selective regret when abstaining on regretful instances.
>
> > **Why do we write yi(Ui) in the definition of Ui... it is a bit confusing**
>
> You are right that this may be confusing. We were assuming the perspective of a practitioner. In this case our intent was to describe the following scenario:
>
> * The practitioner knows $\tilde{y}_i$
>
> * The practitioner does not know the true label $y_i$
>
> This can be written as either $Y_i$ (random variable) or $y(U_i) = y_i \oplus U_i$.
>
> Because $y_i$ is the function of an observed value $\tilde{y}_i$ and a random variable $U_i$, it is itself a random variable. However, we would be happy to use a different notation if you think it would be clearer. Please let us know!

---

### Author Response · Authors · 2024-11-22
**Common Response**

We thank all reviewers for their time and feedback!

We are thrilled that reviewers recognized that our paper tackles a problem that is **"important yet challenging”** [gw9a]. Overall, reviewers regarded this paper as a **“valuable contribution”**[cRDP] through **“rigorous theoretical and empirical analyses”**[cRDP], **“intuitive yet sound mathematical proofs”**[mXWQ], **“convincing experiments”**[mXWQ]. Reviewers also commented on our **“well-written… prose”**[cRDP], and we are pleased we could package these ideas in a paper that the community will find not only interesting but a pleasure to read.

Our rebuttal addresses common feedback among reviewers. We have already addressed some of these in our revision (e.g., missing motivation, nits, typos). We hope to address other questions or concerns over the coming days based on the outcome of our discussion (e.g., motivation for ambiguity metric). We look forward to engaging with everyone over the coming days!

Please let us know if you have any further questions!

---

### Meta-Review · Area_Chair_YAfV · 2024-12-21

**Metareview:**

The paper introduces a novel approach for learning from noisy labels by enabling practitioners to specify a noise model and reverse-engineer pseudo-clean labels for training models.


Strengths

+ Tackles the instance-level noise issue for generating plausible clean datasets

+ Demonstrate improvements in accuracy and uncertainty quantification.

Weaknesses

+ Relies on the practitioner to specify a reasonable noise model, which might be challenging in practice.

+ Limited comparisons with other methods in the literature and insufficient experimental evaluation beyond specific baselines

**Additional Comments On Reviewer Discussion:**

The reviewers are largely in agreement for acceptance, despite some limitations such as the breadth of datasets in the experiments

---

### Decision · Program_Chairs · 2025-01-22

Accept (Poster)